



**Characterization of particle-associated and free-living bacterial and**
**archaeal communities along the water columns of the South China Sea**
Jiangtao Li[a], Lingyuan Gu[a], Shijie Bai[b], Jie Wang[c], Lei Su[a], Bingbing Wei[a], Li Zhang[d] and Jiasong Fang[e,f,g] [*]
[a]State Key Laboratory of Marine Geology, Tongji University, Shanghai 200092, China;
[b] Institute of Deep-Sea Science and Engineering, Chinese Academy of Sciences, Sanya, China;
[c]College of Marine Science, Shanghai Ocean University, Shanghai 201306, China;
[d]School of Earth Sciences, China University of Geosciences, Wuhan, China;
[e]The Shanghai Engineering Research Center of Hadal Science and Technology, Shanghai Ocean University,
Shanghai 201306, China;
[f]Laboratory for Marine Biology and Biotechnology, Qingdao National Laboratory for Marine Science and
Technology, Qingdao 266237, China;
[g]Department of Natural Sciences, Hawaii Pacific University, Kaneohe, HI 96744, USA.
*Corresponding author: jfang@hpu.edu



## Abstract

There is a growing recognition of the role of particle-attached (PA) and free-living (FL) microorganisms in marine carbon cycle. However, current understanding of PA and FL microbial communities is largely on those in the upper photic zone, and relatively fewer studies have focused on microbial communities of the deep ocean. Moreover, archaeal populations receive even less attention. In this study, we determined bacterial and archaeal community structures of both the PA and FL assemblages at different depths, from the surface to the bathypelagic zone along two water column profiles in the South China Sea. Our results suggest that environmental parameters including depth, seawater age, salinity, POC, DOC, DO and silicate play a role in structuring these microbial communities. Generally, the PA microbial communities have relatively low abundance and diversity compared with the FL microbial communities at most depths. Further microbial community analysis revealed that PA and FL fractions generally accommodate significantly divergent microbial compositions at each depth. The PA bacterial communities mainly comprise members of *Actinobacteria* and *γ-Proteobacteria*, together with some from *Bacteroidetes*, *Planctomycetes* and *δ-Proteobacteria*, while the FL bacterial lineages are mostly distributed within *α-*, *γ-Proteobacteria*, *Actinobacteria* and *Bacteroidetes*, along with certain members from *β-*, *δ-Proteobacteria*, *Planctomycetes* and *Firmicutes*. Moreover, there is an obvious shifting in the dominant PA and FL bacterial compositions along the depth profiles from the surface to the bathypelagic deep. By contrast, both PA and FL archaeal communities dominantly consist of Marine Group II (MGII) and Marine Group I (MGI), together with variable minor Marine Group III (MGIII), *Methanosarcinales*, Marine Benthic Group A (MBG-A) and *Woesearchaeota*. However, the pronounced distinction of archaeal community compositions between PA and FL fractions are observed at finer taxonomic level. A high proportion overlap of microbial compositions between PA and FL fractions implies that most microorganisms are potentially generalists with PA and FL dual lifestyle for versatile metabolic flexibility. In addition, microbial distribution along the depth profile indicates a potential vertical connectivity between the surface-specific microbial lineages and those in the deep ocean, likely through microbial attachment to sinking particles.

**Keywords:** particle-attached, free-living, marine microbe, vertical distribution, sinking particles, deep ocean, the South China Sea.



## 1. Introduction

The sinking of particulate organic matter (POM) formed in the photic layer is a fundamental process that transports carbon and nutrient materials from the surface into the usually starved deep ocean, with a significant role in structuring the distributions and activities of marine microorganisms in the dark realm (Azam and Malfatti, 2007; Mestre et al., 2018; Suter et al., 2018). During sinking, the POM is generally colonized and concurrently, decomposed by particle-attached (PA) prokaryotes, releasing dissolved organic matter (DOM) into ambient seawater, fueling the free-living (FL) microbes (Kiorboe and Jackson, 2001; Azam and Malfatti, 2007). It has been revealed that PA and FL microbial populations exhibit different taxonomic composition, physiology and metabolism, corresponding to their lifestyle and ecological behavior. For example, PA bacteria, compared to FL bacteria, are often larger in size (Alldredge et al., 1986; Zhang et al., 2007; Lauro et al., 2009) and metabolically more active (Karner and Herdl, 1992; Grossart et al., 2007). They often maintain higher levels of extracellular enzymes, adhesion proteins and antagonistic compounds, and are capable of degrading high-molecular-weight (HMW) organic compounds (Smith et al., 1992; Crump et al., 1998; Long and Azam, 2001; Mevel et al., 2008; Ganesh et al., 2014). Recently, examination of microbial metagenomes suggests that there are notable differences between PA and FL assemblages in GC content, effective genome size, general taxonomic composition and functional gene categories (Smith et al., 2013). In particularly, some broad key functional gene categories involved in DOM utilization (Poretsky et al., 2010; Rinta-Kanto et al., 2012) and specific functional gene groups linked to successive decomposition of phytoplankton blooms (Teeling et al., 2012) are significantly different, indicating the fundamental differences in survival strategies in relation to potentially available substrates. It is further revealed that PA microbes generally have larger genomes with a variety of metabolic and regulatory capabilities of utilizing compositionally varied organic matter, while the genomes of FL microbes usually are smaller with streamlined metabolic and regulatory functions that enable efficient adaption to oligotrophic conditions (Smith et al., 2013; Yawata et al., 2014; Yung et al., 2016). Phylogenetically, PA and FL lineages generally exhibit different compositions. The PA fraction is relatively enriched in members of *γ-Proteobacteria*, *Verrucomicrobia*, *Bacteroidetes*, *Firmicutes* and *Planctomycetes* (Azam and Malfatti, 2007; Milici et al., 2016; Salazar et al., 2016; Suter et al., 2018), while the FL assemblages are often populated by members of *α-Proteobacteria* (SAR11 clade or *Ca.* Pelagibacter) and *Deferribacteres* (DeLong et al., 1993; Crespo et al., 2013; Milici et al., 2017). However, significantly overlapped compositions of PA and FL microbial communities were also reported in a few studies (Hollibaugh et al., 2000; Ghiglione et al., 2007; Ortega-Retuerta et al., 2013; Rieck et al., 2015; Liu et al., 2018a). Actually, most members of the PA and FL clades are generalists which switch their lifestyles via attachment and detachment to particles (Crespo et al., 2013; Li et al., 2015). As revealed in many marine niches, *α-Proteobacteria*, *γ-Proteobacteria* and *Bacteriodetes* are the major overlapped phyla in both PA and FL microbial fractions (Yung et al., 2016).

Our current knowledge of PA and FL microbial populations largely relies on the upper photic ocean, whereas little information is known from the deep dark ocean, which is the largest biome and accommodates more than half of the ocean's microbes (Aristegui et al., 2009; Salazar et al., 2016). Recently, a number of studies have revealed the PA and FL communities in the bathypelagic waters (Li





et al., 2015; Salazar et al., 2015; Milici et al., 2017; Mestre et al., 2018) or the deepest abyssal and
hadal environments (Eloe et al., 2011; Tarn et al., 2016; Liu et al., 2018a). It is shown that PA and FL
bacterial communities in the deep ocean have clear differences in abundance and composition, in
addition to the detection of novel, unknown prokaryotic taxa. Furthermore, although archaea are a
major component of the marine ecosystem and play significant roles in the degradation of organic
materials (Iverson et al., 2012; Suzuki et al., 2017), PA and FL archaeal communities receive less
attention and little is known about them. Previous limited reports have observed controversial results,
as several studies showed that no obvious differences in archaeal community structures between PA
and FL assemblages (Galand et al., 2008; Eloe et al., 2011; Suzuki et al., 2017), while a clear
separation was found in recent reports (Tarn et al., 2016), with PA archaeal fraction dominated by
Marine Group II (MGII) and Marine Group III (MGII), and FL archaeal fraction by Marine Group I
(MGI) and anaerobic methane-oxidizing archaea (ANME). In brief, it is not well known about the
changes of PA and FL prokaryotes along vertical profiles of water column, from the surface to the
deep bathyal, abyssal and hadal depths.
In this study, we analyzed and compared microbial compositions between PA and FL fractions at
different depths along the vertical profile in the South China Sea (SCS). The SCS is a marginal sea
located in the Northwest Pacific with a maximal depth of approximately 5,380 m (Fig. S1). Our results
reveal diverse and significantly divergent microbial compositions in PA and FL fractions, and obvious
community stratification at different depths along the vertical profiles.
## 2. Materials and Methods
### 2.1 Sample collection and environmental parameter measurements
Seawater samples were collected from two stations, G3 station, depth of 4,039 m at 117° 00.131′ E,
16° 59.947′ N, and J5 station, depth of 4,301 m at 114° 00.209′ E, 13° 59.958′ N, located in the central
deep basin of the SCS during the Open Cruise of R/V *Dongfanghong* II from July 3 to 18, 2014 (Fig.
S1). A Sea-Bird CTD rosette sampler (SBE 911 plus) with 12 L Niskin bottles (Seattle, Washington,
USA) was used to collect seawater from six different depths (50, 200, 1,000, 2,000, 3,000, and 4,000
m) at each station.
Basic environmental parameters of the water column, including depth, salinity, temperature and
dissolved oxygen (DO) were obtained in situ using the conductivity-temperature-depth (CTD) profiler
and a DO sensor during the sampling. Once water samples were collected onboard, about 0.1 L of
seawater was taken immediately for pH measurement with a pH meter (Mettle Toledo Inc.,
Switzerland).
Approximately 8 L of seawater was filtered onboard through a 142 mm precombusted glass fiber
membrane (0.7 μm nominal pore size, Whatman, USA) under a gentle vacuum of <150 mm Hg for
particulate organic carbon (POC) analysis. The membranes were folded and stored at -20 °C until
further analysis. Then about 30 mL of filtered seawater of each sample was taken into 40 mL
precombusted EPA vials and immediately stored at -20 °C for DOC concentration measurement in the



land-based laboratory. About 200 ml filtered seawater at each depth was stored at -20 °C for analysis
of nutrients (NO$_3^-$/NO$_2^-$, dissolved inorganic phosphate and silicate). The remaining seawater was
stored at -20 °C for other analyses.
Approximately 4 L of seawater at each depth was filtered, first through a 47 mm polycarbonate (PC)
membrane of 3.0 μm nominal pore size (Millipore, USA) and subsequently, through a 47 mm PC
membrane of 0.22 μm nominal pore size (Millipore, USA) to collect the particulate-attached and free-
living microbes, respectively (Eloe et al., 2011). The membranes were then frozen at -80°C until
further microbial analysis.
Concentration of POC was determined with a PE2400 Series II CHNS/O analyzer (Perkin Elmer,
USA) (Chen et al., 2008). DOC concentration was measured using a Shimadzu TOC-V Analyzer
(Shimadzu Inc., Japan) (Meng et al., 2017). Nutrients were determined using a Four-channel
Continuous Flow Technicon AA3 Auto-Analyzer (Bran-Lube GmbH, German).
**2.2 DNA extraction**
DNA was extracted from the membranes mentioned above for microbial analysis following the SDS-
based extraction method. Briefly, 800 μl DNA extraction buffer (100 mM Tris-HCl, 100 mM sodium
EDTA, 100 mM sodium phosphate, 1.5 M NaCl, and 1% CTAB) was added into centrifuge tubes
containing the PC membranes. Tubes were frozen-thawed three times by alternating in liquid nitrogen
and a 65°C water bath. Then, 8 μL of 20 mg mL$^{-1}$ proteinase K was added. The solution was incubated
at 37°C for 30 min. Then 80 μL of 10% SDS solution was added, and samples were incubated in a
65°C water bath for 2 h. DNA was extracted by adding water saturated phenol/chloroform/isoamyl
alcohol (25:24:1) and centrifuged at 12,000 ×g for 10 min. The aqueous phase was recovered and
equal volume of chloroform/isoamyl alcohol (24:1) was added again and samples were centrifuged at
12,000 ×g for 10 min. DNA was precipitated with 0.6 volume of cold isopropanol and 0.1 volume of
3M sodium acetate. Samples were incubated at -20°C for 1 h and centrifuged at 12,000 ×g for 10 min.
Finally, DNA pellets were cleaned with 70% cold ethanol, and suspended in 50 μL of sterile deionized
H$_2$O.
**2.3 Pyrosequencing and analysis of 16S rRNA gene sequence amplicons**
The total extracted DNA was quantified with a PicoGreen dsDNA Quantitation Kit (Life
Technologies, USA). The extracted DNA was used as the template for PCR amplification of bacterial
and archaeal 16S rRNA genes with the primers 27F (5′-AGA GTT TGA TCC TGG CTC AG-3′)/533R
(5′-TTA CCG CGG CTG CTG GCA C-3′) containing 10-nucleotide barcodes and Arch344F (5′-ACG
GGG YGC AGC AGG CGC GA-3′)/Arch915R (5′-GTG CTC CCC CGC CAA TTC CT-3′)
containing 8-nucleotide barcodes, respectively. The PCR reactions were held in a thermocycler (Bio-
Rad, USA) at 94°C for 5 min to denature the DNA with amplification proceeding for 25 cycles at
94°C for 50 s, 53°C for 50 s, and 72°C for 50 s. A final extension of 6 min at 72°C was added to
ensure complete amplification. The PCR products purified with the TaKaRa Agarose Gel DNA
Purification Kit (TaKaRa, Japan) were quantified using a NanoDrop ND-1000 device (NanoDrop,





USA) and sent to the 454 FLX Titanium System (Roche, Switzerland) for sequencing.
Downstream analysis of the amplicon reads was performed using QIIME 1.9.1. Reads of low quality
were filtered out by enforcing the following quality control criteria: (1) exclusion of reads with one or
more ambiguous nucleotides; (2) exclusion of reads shorter than 200 bp; (3) exclusion of reads
containing homopolymers of 6 bp and more; (4) exclusion of reads with an average flowgram score of
25 in a quality window of 50 bp. The qualified reads were clustered into operational taxonomic units
(OTUs) based on their sequence similarity (97%), and a representative sequence from each OTU using
the longest picking method was picked for downstream analysis. Taxonomy assignment was
conducted using the RDP classifier against the SILVA 16S rRNA gene database (Version 119).
Chimeric reads were identified and excluded using ChimeraSlayer in the QIIME package after
alignment with PyNAST.
**2.4 Diversity estimators and statistical analyses of microbial communities**
Similarities among different microbial communities were determined using similarity matrices
generated according to the phylogenetic distance between reads (Unifrac distance), and beta diversity
of principal coordinates analysis (PCoA) was computed as components of the QIIME pipeline. The
correlation between the microbial community structures and environmental parameters was analyzed by
canonical correspondence analysis (CCA) and Mantel test. All statistical analyses were performed by R
project (v 3.2.1) using the Vegan and Agricolae packages.
To assess the preference of bacterial lineages for the PA or FL lifestyles, the odds ratio was calculated
for specific clades as below (Ganesh et al., 2014):

180         odds ratio = log 10 (relative abundance in PA fraction / relative abundance in FL fraction)

a positive odds ratio represents higher relative abundance in the PA fraction, while a negative odds ratio
means higher relative abundance in the FL fraction
**2.5 Quantification of microbial 16S rRNA gene and biomass estimation**
Bacterial and archaeal 16S rRNA genes for particle-attached and free-living fractions were quantified
by fluorescence quantitative real-time PCR (Applied Biosystems, UK) with the primer sets
eubac341f/518r (Dilly *et al.*, 2004) and arch344f/519r (Bano *et al.*, 2004), respectively. Amplification
was performed in 20 µl reaction mixture that consisted of 1 µl template DNA (1 to 10 ng), a 0.15 µM
concentration of each primer, and 10 µl of Power SYBR green PCR master mix (Applied Biosystems,
UK) with ROX and SYBR green I. The negative control and gel electrophoresis after each quantitative
PCR experiments were also carried out. For the negative control, only primer dimer with the length of
about 100 bp occurred, while for the samples, just one single and bright band (~ 200 bp) appeared.
Melting curve analysis was performed after amplification and cycle threshold was set automatically
using system 7500 software (1.3). The copy number of 16S rRNA gene was calculated by the average
of triplicate sample. Cell abundance was calculated assuming that every bacterial and archaeal cell



contained 4.08 and 1.71 copies of 16S rRNA gene on average (Lee *et al.*, 2009).
**3. Results**
**3.1 Environmental parameters of the water columns**
Fundamental environmental parameters, including temperature, salinity, pH, DO and POC are listed in
Table 1. In general, they showed similar vertical trends with the normal pelagic ocean. Salinity
increased gradually from ~ 33.84 PSU at 50 m to ~ 34.52 at 200 m and 1,000 m, then maintained at
around 34.6 PSU at greater depths until 4,000 m. DO concentration was the highest (~ 204.5 µM) at
surface water, and decreased gradually to the lowest (~ 83.9 µM) at 1,000 m depth, then increased
gradually from ~ 102 µM at 2,000 m to ~ 113.5 µM at 4,000 m. Nitrite concentrations of the water
columns at all depths were below the detection limit. Concentrations of nitrate, phosphate, and silicate
were continuously increasing from the surface to 1,000 m depth, and then remained at relatively
constant levels (Table 1).
As expected, age of the seawater determined from $\Delta^{14}C_{DIC}$ was youngest at the surface and increased
with depth linearly, varying from about 106 to 1650 years. The upper water layers (50 m and 200 m)
from the two stations had the youngest and nearly the same ages, around 106 years. Ages of 1,000 m
and 2,000 m in G3 station were almost identical, around 1,180 years, and increased to 1,600 years at
3,000 m and 1,750 years at 4,000 m. By contrast, age of 1,000 m in J5 station was ~ 1,310 years, and
remained relatively stable below 1,000 m with the age of about 1,650 years (Table 1). DOC
concentrations ranged from 63.07 to 40.34 µmol/L, with the highest at the surface and lowest at the
deep. However, POC concentrations varied greatly between 0.5 and 2.1 µmol/L and showed great
variations. The POC concentrations were highest at 3,000 m of the G3 station (1.8 µmol/L) and at
1,000 m of the J5 station (2.1 µmol/L) (Table 1).
**3.2 Microbial cell abundances**
The estimated abundances of bacteria and archaea were about $10^6 \sim 10^9$ cells $L^{-1}$ and $10^6 \sim 10^7$ cells $L^{-1}$
, respectively (Fig. 1). The FL bacterial fraction generally accommodated higher cell abundances
(varying from $0.62 \times 10^7$ to $1.65 \times 10^8$ cells $L^{-1}$), several times higher than their corresponding PA
fraction ($1.85 \times 10^6 \sim 1.70 \times 10^9$ cells $L^{-1}$). However, one lower abundance of FL bacterial fraction than
PA fraction was detected in the surface water (50 m) of the G3 station where PA bacterial abundance
was up to $1.23 \times 10^9$ cells $L^{-1}$, two orders of magnitude higher than that of the FL fraction ($1.62 \times 10^7$
cells $L^{-1}$) (Fig. 1a). The upper seawater layers (50 m and 200 m) were also inhabited with the highest
abundance of archaea. FL archaeal fraction had the cell abundances between $1.01 \times 10^6$ and $8.62 \times 10^6$
cells $L^{-1}$, while that of PA archaeal fraction ranged from $1.28 \times 10^5$ to $6.50 \times 10^7$ cells $L^{-1}$. At other
depths, cell densities of archaeal FL fraction varied between $1.01 \sim 3.88 \times 10^6$ cells $L^{-1}$ and $0.74 \sim 8.62$
$\times 10^6$ cells $L^{-1}$ for G3 and J5 stations, respectively. PA archaeal fraction fluctuated between $1.90 \times 10^5$
and $5.54 \times 10^6$ cells $L^{-1}$. Similar to bacteria, the FL archaeal fractions usually showed higher cell
abundances than their PA fractions (Fig. 1b).





### 3.3 Estimation of microbial diversity


Totally 92,041/81,761 and 73,094/97,611 valid sequences of bacterial 16S rRNA gene were obtained
for FL/PA fractions of G3 and J5 stations, respectively. The average valid sequences, including both
PA and FL bacteria were 14, 354 sequences per depth. Based on the 97% similarity, these FL and PA
bacterial sequences were defined into a total of 6,666 operational taxonomic units (OTUs). The
number of OTUs in the FL and PA bacterial fractions at each depth ranged from 214 to 1,470 (Table
S1). Correspondingly, 50,736/41,719 and 44,456/38,333 archaeal sequences were determined for
FL/PA archaea fractions of G3 and J5 stations. Attempt to determine PA archaeal sequence from 3,000
m depth of G3 station and 4,000 m depth of J5 station failed because of technical reasons. The average
number of archaeal sequences (including PA and FL archaea) were 7,966 sequences per depth. A total
of 1,071 archaeal OTUs were defined and the number of OTUs for the FL and PA archaeal fractions
varied from 82 to 275 (Table S2).
Shannon's diversity (H) and Chao1 were calculated to estimate microbial diversity of both PA and FL
fractions at all depths (Fig. 2 and Fig. S2). In most cases, the H indices of the bacterial FL fractions
were always higher than their PA counterparts at each depth (Fig. 2). H index of FL and PA bacterial
fractions gradually increased from 50 to 1,000 m, decreased from 1,000 to 2,000 m, and increased
again from 2,000 to 4,000 m (Fig. 2a). Similar to bacteria, FL archaea had higher H index values than
the PA fraction. The H index was usually the lowest at the surface, increased to the highest value at
200 m or 1,000 m and decreased continuously into the deep (Fig. 2b). Chao1 index showed similar
variation trends for both PA and FL microbial fractions (Fig. S2).
PCoA analysis revealed that there were significant differences in bacteria and archaea community
structures over the depth profiles and between the FL and PA fractions. Overall, three groups were
distinguished, the surficial 50 m group, the FL group, and the PA group (Fig. 3). One incompact group,
consisted exclusively of samples at 50 m depth, separated the microbes in the surface from those in the
rest of the water column of both stations, irrespective of microbial lifestyles (FL or PA). However, the
other two groups were separated mainly based on the FL and PA lifestyles. It is interesting to note that
the FL bacterial samples clustered into one group where samples were further partitioned with respect
to depth (Fig. 3a). Canonical correspondence analysis (CCA) showed that fundamental environmental
parameters including depth, DO, salinity, seawater age, DOC and POC concentration, and silicate
exerted potential impact on variations of FL and PA microbial communities along the water column
(Fig. 4, Fig. S3). Mantel test further indicated that all those factors , except POC concentration ($P$
=0.164), were the statistically significant variables associated with variation of PA and FL fractions ($P$
=0.001).

### 3.4 Taxonomic compositions of the PA and FL bacterial and archaeal fractions


Taxonomic compositions of FL and PA bacterial fractions and their relative abundances are presented
in Fig. 5. At phylum level, bacterial sequences were mainly assigned into *Proteobacteria* (*α-, β-, γ-,*
and *δ-*), *Actinobacteria*, *Cyanobacteria*, *Planctomycetes*, *Bacteroidetes*, *Marinimicrobia* (SAR406
clade), *Chloroflexi*, *Firmicutes*, *Gemmatimonadetes*, *Gracilibacteria* and *Verrucomirobia*. The taxa at



family level with relatively high abundances on average in either PA or FL fraction were further shown
in Fig. 6.
It is clear that *α*- and *γ-Proteobacteria* were the dominant lineages in both the FL and PA fractions at
nearly all depths. In most cases, the sum of *α*- and *γ-Proteobacteria* accounted for ~ 40% to nearly
90%. Moreover, their relative abundances in different PA and FL fractions and different stations also
varied widely. Within the *α-Proteobacteria*, the dominant families included *Methylobacteriaceae*,
*Phyllobacteriaceae*, *Rhodobacteraceae* and *Erythrobacteraceae* (Fig. 6). Members of the families
*Methylobacteriaceae* and *Erythrobacteraceae* occurred commonly in both fractions at almost all
depths but usually with higher proportions in PA fractions. The family *Rhodobacteraceae* occurred
commonly in both fractions at every depth (1 % ~ 20%), while the *Phyllobacteriaceae* was dominantly
distributed in the PA fraction of 2,000 m depth of J5 station with > 60% proportions. In addition,
another important lineage within *α-Proteobacteria* is SAR11 clade (now named as *Pelagibacterales*)
(Grote et al., 2012). It was clearly revealed that SAR11 clade showed relative higher abundances in FL
fractions than PA fractions. Moreover, at depths above 1000 m, SAR11 clade had a far higher
proportion than the deep ocean and the maximum levels occurred at 200 m depth (20% ~ 24%) (Fig. 6,
Table S1). *γ-Proteobacteria* is another lineage with the highest abundance overall. Its relative
abundances change significantly with depths and in different fractions. The minimum abundances
were only 1% ~ 5%, while the maximum were up to 73% ~ 80% (Fig. 5 and Table S1). Moreover, G3
station generally had higher *γ-proteobacteria* proportions than that of J5 station on average. As shown
in Fig. 6, although sequences of *γ-Proteobacteria* were classified into multiple families, actually only
two families *Alteromonadaceae* and *Pseudoalteromonaodaceae* exhibited dominant prevalence in the
bacterial populations. The *Pseudoalteromodaceae* populated predominantly the PA fractions in 50 m
and 200 m depths (66% ~ 75%), while the *Alteromonadaceae* mainly dominated the PA fractions in
the deep water, particularly at 2,000 m and 3,000 m depths. *δ-Proteobacteria* also had a common
distribution in both fractions of all depths, usually accounting for less than 10% proportions in most
samples (Fig. 5), and SAR324 clade members contributed significantly to the dominance of the *δ-*
*Proteobacteria* (Fig. 6). *Actinobacteria* and *Cyanobacteria* were abundantly distributed only in the
surficial 50 m depth, and by sharp contrast, their proportions in other depths were less than 5%. Other
bacterial lineages which had a wide distribution in all depths but only with minor abundances in both
fractions included *Planctomycetes*, *Bacteroidetes*, *Marinimicrobia* (SAR406 clade), *Chloroflexi*, *β-*
*Proteobacteria*, *Firmicutes*, *Gemmatimonadetes* and *Verrucomicrobia* (Fig. S4).
Majority of archaeal amplicons were mainly fallen into several uncultured taxonomic lineages (Fig. 7
and Fig. S5). Both FL and PA archaeal fractions at all depths were principally populated by Marine
Group I (MGI) of the *Thaumarchaeota* and Marine Group II (MGII) of the *Euryarchaeata*. Members
from MGI and MGII lineages generally contributed more than 80% relative abundances in their
respective clone libraries. MGI was always one of the most abundant clades along the vertical profiles
except in the topmost FL and PA fractions. Within the MGI group, only a small part of members were
annotated into the cultured genus *Nitrosopumilus* and *Candidatus* Nitrosopelagicus, while the majority
of them fell into those uncultured subclades (Table S2). MGII clade exhibited a wide distribution
along the water columns, and it usually accounted for the large proportions in both archaeal size
fractions. The photic layer (~ 50 m depth) contained the highest abundances of MGII clade,
particularly in FL fractions with up to ~ 80% proportions. By sharp contrast, the lowest abundances of
MGII occurred at 2,000 m (G3 station) and 3,000 m (J5 station) depths, making up <20% percentages.



The third most abundant clade overall is Marine Group III (MGIII) of the *Euryarchaeata*. MGIII
representatives were mainly dispersed in the FL fractions with 5% ~ 18% abundances, while they were
absent from most of the PA fractions. The order *Methanosarcinales* of *Euryarchaeata* was detected
commonly in most PA fractions, but it had the higher abundance only in the upmost 50 m depth (~
29.7%) (Fig. 7). Another sample accommodating relatively much *Methanosarcinales* was the PA
faction of 3,000 m in J5 station with 9.1% proportion. Within the *Euryarchaeata*, another clade of
methanogens, *Methanobacteriales*, was also detected from both size fractions but with low relative
abundances (<5%) (Fig. 7, Fig. S5, Table S2). In addition, other archaeal lineages included
*Woesearchaeota* (formerly known as the DHVEG-6 group), Miscellancous Crenarchaeotic Group
(MCG, now named as *Bathyarchaeota*), the *Halobacteriales* of the *Euryarchaeata* and Marine Benthic
Group A (MBG-A) of the *Thaumarchaeota*. They just provided a limited contribution to archaeal
populations (Fig. S5).

### 324   3.5 Bacterial preference to PA or FL lifestyles

Odds ratio was used to assess the preference of bacterial taxonomic lineages to the PA or FL lifestyle.
A positive odds ratio indicates PA preference or higher abundance in the PA fraction, while a negative
value suggests FL preference or higher abundance in the FL fraction. The bacterial lineages
dominating the PA fractions come exclusively from *α-* and *γ-Proteobacteria* (Fig. 6). At family level,
the dominant clades comprised of the *Phyllobacteriaceae*, *Methylobacteriaceae*, *Erythrobacteraceae*,
*Rhodobacteraceae* (*α-Proteobacteria*), and *Pseudoalteromonadaceae*, *Alteromonadaceae* (*γ-*
*Proteobacteria*) (Fig. 6) and they show a clear preference to PA lifestyle at different depths (Fig. 8).
Except for these prevalent families, there is a wide range of lineages also showing preference to
particle-attached lifestyle but with relatively low abundance (Fig. 6 and Fig. 8). These minor lineages
are mainly populated by the families *Oceanospirillaceae* and *Alcanivoracaceae* (*γ-Proteobacteria*),
*Sandaracinaceae* and *Bdellovibrionaceae* (*δ-Proteobacteria*), *Burkholderiaceae* (*β-Proteobacteria*),
*Saprospiraceae* (*Bacteroidetes*), *Planctomycetaceae* and *Phycisphaeraceae* (*Planctomycetes*),
SAR406 clade (*Marinimicrobia*), *Cryomorphaceae* and *Flavobacteriaceae* (*Bacteroidetes*),
*Propionibacteriaceae*, *Nocardioidaceae* and *Corynebacteriaceae* (*Actinobacteria*).
The predominant lineages of FL fractions mainly consisted of members of *Actinobacteria*,
*Cyanobacteria*, *Bacteroidetes*, α- and δ-*Proteobacteria*, as shown in Fig.5. At family level, the
phylogenetic lineages with showing a FL preference are mainly populated by the families OM1 clade
and Sva0996 marine group (*Actinobacteria*), SAR324 clade and *Nitrospinaceae* (*δ-Proteobacteria*),
*Cyanobacteria*, *Comamonadaceae* (*β-Proteobacteria*), *Erythrobacteraceae*, SAR11 clade,
*Methylobacteriaceae*, *Bradyrhizobiaceae*, *Rhodobacteraceae*, *Hyphomonadaceae* (α-Proteobacteria),
*Phycisphaeraceae* and *Phycisphaeraceae* (*Planctomycetes*), SAR406 clade, *Saprospiraceae*,
*Chitinophagaceae*, *Cryomorphaceae*, *Flavobacteriaceae*, *Flammeovirgaceae* (Bacteroidetes) (Fig. 8).
However, compared with counterparts of PA fractions, their abundances in FL fractions are low
without absolute dominance.




## 4. Discussion

### 4.1 Comparison of microbial abundance and diversity between PA and FL fractions

PA bacterial and archaeal fractions show generally lower abundance and taxonomic richness than their FL counterparts and constitute a small fraction of the total abundances. Our results are consistent in principle with previous reports on various pelagic environments, in either the euphotic zone, twilight or the dark deep ocean (Turley and Stutt, 2000; Simon et al., 2002; Ghiglione et al., 2007; Rieck et al., 2015). However, in some eutrophic and notably particle-rich marine ecosystems, for example, marine snow or estuaries, PA bacterial fractions were present in higher local concentrations and greater diversity than FL bacteria (Caron et al., 1982; Karner and Herndl, 1992; Turley and Mackie, 1994; Garneau et al., 2009). In upper photic zone, PA bacterial abundance and their contribution to total bacterial biomass are highly variable, and depend largely on the quantity and quality of suspended organic particles (Cammen and Walker, 1982; Simon et al., 2002; Doxaran et al., 2012). This is indeed the case in the South China Sea. As shown in Fig. 1, at 50 m and 200 m depths of G3 station, PA bacterial abundances outnumbered FL bacteria by nearly 2 ~ 100 times, whereas J5 station has an opposite trend. However, as shown in Table 1, these two stations have almost the same environmental parameters, particularly in POC concentrations. One possibility may be that G3 and J5 have different POC compositions, attributable to different origins of organic matter. Although bacteria attaching to particles are of relatively lower abundance compared to free-living cells in the pelagic ocean, they are consistently metabolically more active with higher extracellular enzymatic activities (Karner and Herndl, 1992) and cell-specific thymidine incorporation rates (Turley and Mackie, 1994; Turly and Stutt, 2000). Therefore, PA bacteria often play a comparable role to free-living bacteria in hydrolysis or decomposition of marine organic matter, biomass production and carbon cycling (Griffith et al., 1994; Turly and Stutt, 2000; Liu et al., 2015). The decline of bacterial abundance and richness along the depth profile is largely owing to the gradual decreasing availability of usable organic carbon (Smith, 1992; Turly and Stutt, 2000; Jiao et al., 2014). In contrast, archaea are commonly much lower in cell abundance and community diversity compared with their bacterial counterparts at the same depths (Fig. 1-2 and Fig. S2). The relative abundance of archaeal populations in total prokaryotes increases gradually with depth, indicative of a potential rising impact on biogeochemical cycle in marine environments. In addition, pronounced distinction in microbial community structures of PA and FL assemblages were observed along the depth profile, which were well supported by results of statistical analyses (Fig. 3). It is expectable that PA fraction differs taxonomically from FL fraction, considering their discrepant activity patterns for survival. Related discussions are shown below.

### 4.2 Environmental factors potentially shaping microbial community structure

Several environmental parameters were supposed to play a pivotal role in structuring microbial communities of seawater. Depth, together with age and salinity of water mass, are a key subset of environmental drivers (Fig. 4). Recent studies have shown that microbial populations in the meso-/ bathypelagic ocean are largely dissimilar to those of the epipelagic zone (Salazar et al., 2015; Milici et al., 2017; Liu et al., 2018a), indicative of a crucial environmental selection process exerted by depth.



In our study, PCoA analysis revealed that PA and FL fractions from the surficial zone (50 m) were
clustered into a separate but relatively loose group distant from other depths (Fig. 3), indicative of the
influence imposed from depth in shaping microbial community structures. Several bacterial lineages,
including *Cyanobacteria*, *Actinobacteria*, *δ-Proteobacteria*, *Marinimicrobia* (SAR406 clade) and
*Firmicutes* with distinct distributing stratification contribute to this dissimilarity. *Cyanobacteria* and
*Actinobacteria* belong to typical phototrophs (Mizuno et al., 2015) and they are prevalently distributed
in euphotic zones. By contrast, δ-proteobacterial SAR324 clade, as shown in our results, are primarily
found in mesopelagic waters (200 ~ 1,000 m) (Fuhrman and Davis, 1997; Wright et al., 1997).
SAR406 clade has a ubiquitous distribution across diverse marine niches, however, its high abundance
always occurs within the mesopelagic zones, ~ five times or higher than in surface ocean (Yilmaz et
al., 2016). Archaeal population components also reflect the impaction of depth. Euphotic zones hold
less abundant thaumarchaeotal MGI and more euryarchaeotal *Methanosarcinales* and *Woesearchaeota*
(Fig. 7), while marine thaumarchaeotal groups are more abundant in meso- and bathypelagic waters
(Karner et al., 2001; Mincer et al., 2007; Varela et al., 2008). In addition, Salazar et al. (2016) found
that sampling depth appears to have a more direct impact on free-living bacterial communities. Our
results are highly consistent with this observation in that FL bacterial fractions from the same depth
grouped together irrespective of their sampling locations (G3 or J5 station) (Fig. 3a).
DO concentration is observed to strongly affect particle flux and particle transfer efficiency from
euphotic zone to the deep sea since remineralization of organic particles appears to be oxygen-
dependent (Laufkotter et al., 2017; Cram et al., 2018). It is considered as one of the best subsets of
environmental variables for shaping the compositions of particle-attached bacterial assemblages
(Salazar et al., 2016). Some taxonomic lineages are directly affected by oxygen. For example, a most
recent study found that oxygen is one of the key factors driving the distribution and evolutionary
diversity of *Woesearchaeota* (Liu et al., 2018b). POC and DOC can be substrates for both PA and FL
communities, respectively (Azam and Malfatti, 2007; Zhang et al., 2016; Liu et al., 2019). However,
POC concentration in the present study is not statistically significantly correlated with either bacterial
or archaeal community abundances ($P > 0.05$). We hypothesize that the quality rather than the quantity
of POC imposes a decisive influence on microbial populations, especially in the deep, dark ocean.
During the POC sinking from surface through the water column, the labile organic matter becomes
increasingly decomposed, while the more refractory material remains and resists degradation (Simon
et al., 2002). In such cases, utilization of refractory POC by microorganisms depends on the quality of
POC. Among common nutrients, silicate exhibited statistically significant correlation with microbial
distributions (Fig. S3), and this is out of our expectation because the SCS generally shows N- or P-
limit in phytoplankton production (Wu et al., 2003; Chen et al., 2004). However, recent research found
that near the sampling site of this study, there is a clear silicon deficiency in the euphotic zones
shallower than 75 m (Huang et al., 2015), which directly influences the diversity and biomass of
phytoplankton, and consequently, the quantity and quality of POM transported to the deep along the
vertical water columns, and finally exerts a potential impact on microbial communities. Actually,
microbial community structure and their distribution along the water column profile are a
comprehensive combination impacted by multiple environmental variables.



### 4.3 Specialist or generalist for PA and FL lifestyle: clues from bacterial community compositions

It was suggested that PA and FL bacterial fractions accommodated different phylogenetic compositions along the depth profiles (Fig. 3), consistent with previous reports in various marine niches (Acinas et al., 1997; Moeseneder et al., 2001; Ghiglione et al., 2009; Salazar et al., 2015). However, in most cases, taxonomic compositional disparity between the two filtration fractions does not seem much apparent at phylum level (Fig. 5). Actually, a few studies also confirmed that at high taxonomic ranks, bacteria show conserved lifestyles either in association with particles or as free-living microorganism (Eloe et al., 2011; Salazar et al., 2015; Liu et al., 2018a). The pronounced contrast in population compositions of the two filtration fractions was unveiled only at greater taxonomic level and a considerable number of phylogenetic taxa exhibited different preferences to PA or FL lifestyles. As shown in Fig.5 and Fig.6, as the most abundant members, *α- and γ-Proteobacteria* occurred prevalently in both filtration fractions, but at the family level, most of predominant bacterial lineages of PA and FL fractions were significantly divergent, indicating their preference to different microhabitats shaped by organic particles and environmental parameters. The dominant lineages in PA fractions were mainly associated with the families *Pseudoalteromonadaceae* and *Alteromonadaceae* within *γ-Proteobacteria*, and the *Methylobacteriaceae* within *α-Proteobacteria*. These *γ-proteobacterial* members are usually retrieved from diverse marine habitats as the typical PA clades, and they are believed to have the abilities to degrade/utilize HMW organic compounds with higher nutrient requirements (DeLong et al., 1993; Crespo et al., 2013). The adhesion to particles could make them increase nutrients acquisition and avoid the nutrient-depleted conditions (Crespo et al., 2013). By contrast, members of *α-Proteobacteria* are rarely reported as the dominant lineages of PA fraction or particle-attached preference (Crespo et al., 2013; Rieck et al., 2015; Suzuki et al., 2017), which is inconsistent with our results revealing α-proteobacterial lineages frequently prevail as PA members. Further phylogenetic assignment revealed that the majority of α-proteobacterial PA members exclusively belong to the genus *Methylobacterium* which are strictly aerobic, facultatively methylotrophic bacteria, and can grow on a wide range of carbon compounds (Green, 2006). They probably benefit from the particle-attached lifestyle, making their high requirements for organic matters easily to achieve. Compared with bacterial PA counterparts, FL bacterial communities are more diverse, and dominant populations are scattered in more phylogenetic taxa with relatively homogeneous proportions. Among the predominant lineages, the actinobacterial OM1 cade and cyanobacteria dominantly govern the upper surficial waters (Fig. 6), likely attributed to their phototrophic behaviors. Although actinobacteria are recognized as ubiquitous members of marine bacterioplankton (Giovannoni and Stingl, 2005), they are scarcely reported with predominance (Milici et al., 2016a). Recently, Ghai et al. (2013) revealed the OM1 clade members possess the smallest cell sizes with streamlined genome, representing a typical adaption to oligotrophic condition (Giovannoni et al., 2014) which well agrees with the oligotrophic environments in the SCS (li). Other predominant FL lineages include α-proteobacterial SAR11 clade, δ-proteobacterial SAR324 clade, and *Marinimicrobia* (SAR406 clade), all usually being the most ubiquitous free-living bacterial lineages and dominantly distributed in epi- and mesopelagic zones (Grote et al., 2012; Tarn et al., 2016; Yilmaz et al., 2016; Milici et al., 2017; Liu et al., 2018a). Genomic information underlines that although these clades have a flexible metabolism utilizing multiple hydrocarbon compounds, they generally lack of carbohydrate-active enzyme genes for the attachment to and the degradation of particulate organic matter (Peoples et al., 2018), consistent with their preference to free-living lifestyle rather than





particle-attachment (Eloe et al., 2011; Salazar et al., 2015; Tarn et al., 2016).
In addition to those predominant lineages mentioned above, there are a couple of bacterial taxa
showing evident PA or FL preferences. At ~ family level, these PA- or FL-preferred taxa are well
hinted by their odds ratio between PA and FL fractions. These bacterial lineages are characterized by
low abundances or occasional occurrence in water columns (Fig. 6) but high odds ratio (absolute
value) (Fig. 8), indicating their strong preferential divergence in the two size fractions. As shown in
Fig. 8, such families with PA preference were mainly derived from the phyla/classes *Actinobacteria*
and *γ-Proteobacteria*, together with several families from *Bacteroidetes*, *Planctomycetes* and *δ-*
*Proteobacteria*, while FL-preferred lineages are mostly distributed within *α-*, *γ-Proteobacteria*,
*Actinobacteria* and *Bacteroidetes,* along with certain groups of *β-*, *δ-Proteobacteria*, *Planctomycetes*
and *Firmicutes*. The majority of these lineages are recorded consistently about their PA- or FL
preferences in previous studies, and commonly possess the ability to hydrolyze and utilize complex
carbon sources. Although their abundance is low, these minor populations can still effectively
influence local microhabitats because of their high specificity for organics. In contrast, there are still
some populations which are scarcely reported. For example, Sva0996 marine group, an actinobacterial
group, is retrieved occasionally from marine sediments and upper ocean (Bano and Hollibaugh, 2002;
Wang et al., 2018). Orsi et al. (2016) first found this group prefers to free-living lifestyle in upper
seawater and have the ability to assimilate phytoplankton-derived dissolved protein. Our present
results suggest that Sva0996 group are flexible to adapt PA or FL lifestyles at the surface seawater
because two lifestyles occur concurrently. Moreover, the distribution of Sva0996 group is not
restricted only in upper photic ocean, and they can survive in meso- and bathypelagic seawaters with
the significant preference for free-living lifestyle (odds ratio for FL-preference is up to 3.93).
However, nothing is available to elaborate the selection between PA and FL lifestyles due to lack of
pure culture or their genome information.
A high proportion of bacterial lineages are revealed to co-occur in both PA and FL fractions. At OTU
level, more than 1/3 of total OTU numbers (2402 out of 6964 OTUs) are shared by PA and FL
fractions (Fig. 9). Phylogenetically, these PA/FL-shared OTUs are mainly fallen into *α-*, *γ-*, *δ-*
*Proteobacteria*, *Planctomycetes*, *Bacteroidetes* and *Actinobacteria*. Moreover, taxonomic components
of PA/FL-shared OTUs at different levels are primarily similar to those of OTUs retrieved exclusively
from PA fractions or FL fractions (Table S1, Fig. 9), indicating that a considerable amount of bacterial
lineages potentially have PA and FL dual lifestyle strategies (Bauer et al., 2006; Gonzalez et al., 2008).
On the one hand, a few lineages such as *Flavobacteriaceae*, *Planctomycetaceae*, *Rhodobacteraceae*,
*Erythrobacteraceae*, *Burkholderiaceae*, *Nitrospinaceae*, SAR324 clade, *Alteromonadaceae*,
*Pseudomonadaceae* and *Salinisphaeraceae* co-occur in PA and FL fractions at least at one of the same
depths with approximately equivalent abundances. In such cases, their odds ratios are close to zero or
minor range, indicating that bacteria are able to employ two different survival strategies at the same
time. On the other hand, some taxa including the families Sva0996 marine group, *Flavobacteriaceae*,
*Phycisphaeraceae*, *Rhodobacteraceae Methylobacteriaceae*, *Erythrobacteraceae*,
*Pseudoalteromonadaceae*, *Halomonadaceae* and *Moraxellaceae*, show divergent preferences to PA or
FL lifestyles at different depths or different locations. This is clearly evident by the shift or conversion
of their odds ratios at different depths along the vertical profiles of water column (Fig. 9), indicative of
their different adaption tactics to different environments. One possible explanation is that most of the
marine bacteria are generalists with dual life strategies (Bauer et al., 2006; Gonzalez et al., 2008), and



able to grow in suspension as well as on particles (Lee et al., 2004; Grossart et al., 2006, 2010). For
instance, PA bacteria must be capable of surviving freely in the water column to migrate and colonize
new organic particles (Ghiglione et al., 2007; Crespo et al., 2013). Bacterial populations may switch
their lifestyles between free-living and particle-attachment, depending on substrate availability and the
surrounding chemical triggers (Grossart, 2010; D'Ambrosio et al., 2014). To date, one exception, the
genus *Scalindua* in the *Planctomycetes* phylum, which is a known marine chemoautotroph involved in
anammox, is exclusively observed in FL fractions in previous studies (Fuchsman et al., 2012; Ganesh
et al., 2014; Suter et al., 2018). However, it is absent from our water columns.

### 4.4 Archaeal community preferences to PA and FL lifestyles

Samples of PA and FL archaeal fractions were also separated into different groups by statistical
analysis (Fig. 3b), indicating their phylogenetically different community structures. However, because
most of OTUs belonged to uncultured archaeon, it is impossible to assign them into taxonomic
lineages at finer level. Thus, the distinction of archaeal population compositions between PA and FL
fractions was unnoticeable (Fig. 7). The MGI and MGII are the most abundant taxa in both PA and FL
archaeal fractions. The MGI thaumarchaea are one of the most abundant and cosmopolitan
chemolithoautotrophs in the dark ocean (Karner et al., 2001) and responsible for much of the ammonia
oxidation in this environment for their common metabolism of aerobic ammonia oxidation.
Corresponding to their autotrophic metabolism, MGI generally exhibit free-living preference and are
the prevalent archaeal taxa in free-living fractions below euphotic zone (Smith et al., 2013; Salazar et
al., 2015; Tarn et al., 2016). However, different from our results, a few studies showed that MGI
dominated both the PA and FL archaeal populations and no obvious distinction was observed in
abundance and ecotype of MGI (Eloe et al., 2011; Jin et al., 2018). To date, only a few pure cultures of
marine MGI, small rods with a diameter of 0.15~0.26 μm and a length of 0.5 ~ 1.59 μm and no
flagella were observed (Könneke et al., 2005; Qin et al., 2014), suggesting that their occurrence in PA
fraction is not caused by pore size of filter to fractionate different assemblages. One possibility is that
decomposition of organic particles continuously releases ammonia and MGI can easily acquire high
concentrations of ammonia by attaching to particles, especially in oligotrophic area. Recent studies
provide another explanation to particle-attached MGI that some MGI cultures are obligate mixotrophy
that rely on uptake and assimilation of organic compounds (Alonso-Sáez et al., 2012; Qin et al., 2014).
In such case, PA lifestyle is in favor of their nutrient requirements. MGII have a wide distribution in
the open ocean and as shown in our results, they are the dominant archaeal community generally
within the upper euphotic zone (Massana et al., 2000; Martin-Cuadrado et al., 2015). Recently, they
have been found, however, to be also abundant in deep-sea waters (Baker et al., 2013; Tarn et al.,
2016; Liu et al., 2018a), showing a wider adaption to diverse marine habitats in addition to the photic
zone. MGII are thought to be heterotrophs, and have the ability of degrading proteins and lipids
(Iverson et al., 2012; Orsi et al., 2015). Metagenomes revealed a number of genes encoding cell
adhesion, degradation of high molecular weight organic matter and photoheterotrophy (Rinke et al.,
2019; Tully et al., 2019), evidencing their potentiality to utilize organic particles as important growth
substrates. All these findings imply MGII's preference to particle-attached lifestyle, and they are
frequently detected from PA fractions in size-fractionated studies (Iverson et al., 2012; Orsi et al.,
2015; Tran et al., 2016). However, in a few studies including our present study, MGII are also
identified as the dominant archaeal components from FL fractions, with equal or even more abundance



than PA fractions (Fig. 7). Further studies confirm that genome contents and populations of free-living
MGII are distinct from those of particle-attached MGII (Orsi et al., 2015; Rinke et al., 2019),
suggesting their metabolic evolution and adjustment to niche partitioning. In addition, MGIII also
occurred commonly in both fractions (Fig. 7). MGIII are usually retrieved as minor components of
deep mesopelagic and bathypelagic communities (Galand et al., 2009; Tarn et al., 2016). Like MGII,
to date no cultured representative of MGIII leads to little is known about their ecological and
physiological characteristics. Function prediction from metagenomes suggest that MGIII are aerobic
(or facultative anaerobic), motile, and heterotrophic, and potentially can utilize lipid, proteins and
polysaccharides as major energy source (Martin-Cuadrado et al., 2008; Haro-Moreno et al., 2017).
Recently, a novel lineage of MGIII genomes preferring to live in the photic zone was recovered,
consistent with previous few studies and our present results in which MGIII populations are obtained
from the euphotic zone with a considerable abundance (Galand et al., 2009, 2010). Moreover, recent
findings also indicate that MGIII are inclined to be attached to other microorganisms (particle-attached
preference) and only sporadically be released to the surrounding environments (free-living lifestyle)
(Haro-Moreno et al., 2017).
In addition, there are several other archaeal lineages with remarkable differences in abundance
between PA and FL fractions. The order *Methanosarcinales* and *Methanobacteriales*, affiliated to the
phylum *Euryarchaeota* and retrieved exclusively from PA fractions (Fig. 7), belong to strictly
anaerobic methanogens. Their preference to particle-attached lifestyle in water column environments
is intelligibly convinced. Within normal water column, seawater is oxic in spite of low oxygen
concentration and only on or inside the particles where heterotrophic microbes attach and digest
organic matter using oxygen as electron acceptor, local anoxic niches are developed with the
exhaustion of ambient oxygen and become suitable for the survival of methanogens. Members of the
*Woesearchaeota* were abundantly derived from the PA fraction of the upper seawater. In marine
environments, *Woesearchaeota* are distributed restrictively in marine sediments (Lipsewers et al.,
2018) or deep-sea hydrothermal vents (Takai et al., 1999), and are scarcely detected from pelagic
seawater masses. Recent studies suggest that woesearchaeotal lineages are mostly retrieved from
anoxic environments (Castelle et al., 2015; Liu et al., 2018b). Moreover, genomic metabolic analysis
indicates *Woesearchaeota* have an anaerobic heterotrophic lifestyle with conspicuous metabolic
deficiencies (Probst et al., 2017; Liu et al., 2018b), implying a potential syntrophic or mutualistic
partnership with other organisms (Castelle et al., 2015; Liu et al., 2018b). It is further demonstrated
that *Woesearchaeota* tend to co-occur with typical anaerobic methanogens from the *Methanomicrobia*
and *Methanobacteria* constituting a potential consortia (Liu et al., 2018b). In our present results, at
several depths, the *Methanosarcinales* of the *Methanomicrobia* and the *Methanobacteriales* of the
*Methanobacteria*, together with *Woesearchaeota*, were detected concurrently, implying to a large
extent their potential syntrophic partnership.

**4.5 Potential vertical connectivity of microbial populations along the depth profile**

Microbial distribution at different depths to a certain extent implicates their potential vertical
connectivity along the water column profile. It has been suggested that the sinking of organic particles
formed in upper euphotic zone is a main vector in transferring prokaryotes from the surficial ocean to
deep waters (Mestre et al., 2018). Those surficial lineages, usually belonging to putative



photosynthetic/photoheterotrophic, Bchl a-containing microorganism or strict epipelagic/euphotic
inhabitants, are reliable indicators to hint their downward transportation if they are detected from
meso- or bathypelagic waters. For example, cyanobacteria are typical photosynthetic bacteria and their
distribution is thought to be confined to the euphotic zone, with commonly observed maximum depths
of about 150 ~ 200 m. In the present study, however, cyanobacterial lineages were retrieved
throughout the whole water column (Fig. 5 and Fig. 6), especially at 4,000 m depth where
cyanobacteria account for nearly 12% of the PA communities. Although a recent study revealed that
cyanobacteria can dominate the deep continental subsurface microbial communities with the potential
for a hydrogen-based lithoautotrophic metabolism instead of photosynthesis (Puente-Sanchez et al.,
2018), these indigenous deep cyanobacteria were classified into the genera *Calothrix*, *Microcoleus* and
*Chroococcidiopsis*, phylogenetically different from those prevailing in our study (*Prochlorococcus*,
*Synechococcus*). Jiao et al. (2014) observed substantial *Prochlorococcus* populations at 1,500 m depth
in the South China Sea, and suggested that multiple physical processes, including internal solitary
waves and mesoscale eddies were responsible for the occurrence of these "deep *Prochlorococcus*".
However, in our study area, ages of seawater increase gradually from the surface to the deep along the
water column profile in a normal time sequence (Table 1), refuting this possibility. Thus, a reasonable
postulation here is that the sinking particles function as vectors and convey cyanobacteria attaching on
particle surfaces from epipelagic zone into deep-sea waters. Likewise, members of the family
*Erythrobacteraceae*, which are largely represented by OTUs within the genus *Erythrobacter*, are also
present abundantly in both PA and FL fractions at 4,000 m depth (Fig. 6). *Erythrobacter* spp. belong to
putative Bchl a-containing, aerobic anoxygenic photoheterotrophic bacteria and are thought to be
distributed only in the euphotic upper ocean (Kolber et al., 2000; Koblížek et al., 2003). SAR11 clade,
are potentially photoheterotrophic (Gomez-Pereira et al., 2013; Evans et al., 2015) and ubiquitous in
global photic zones as one of the most abundant bacteria (Morris et al., 2002). We observed that
members of SAR11 clade are distributed across the whole water columns, especially in mesopelagic
aphotic depths with relatively high proportions. Other lineages specializing in inhabiting surface
seawater but was also retrieved from the deep ocean include γ-proteobacterial SAR86 clade, SAR116
clade of marine Roseobacter and SAR202 clade within *Chloroflexi*. The majority of the OTUs within
these "surface lineages" have been retrieved from the meso-/bathypelagic ocean and can be traced
back simultaneously to those present in surface waters, suggesting their potential origin from the upper
epipelagic zones.
**5. Conclusions**
In this study, we systematically compared bacterial and archaeal community structures within two
different filtration fractions representing particle-attached and free-living lifestyles at different depths
in the South China Sea. As revealed in previous studies, whatever bacteria or archaea, the FL fractions
usually show higher cell abundance and diversity than their PA counterparts at most depths. A set of
environmental factors including depth, salinity, seawater age, DOC, POC, DO and silicate are
considered playing important roles in structuring PA and FL microbial communities along the depth
profile. On the one hand, as the result of adapting to different organic substrates available, PA and FL
fractions generally accommodate significantly divergent microbial compositions at each depth. At fine
taxonomic levels, a considerable number of microbial lineages exhibit pronounced preferences to PA
or FL lifestyles, also with distinct distributing stratification along the depth profile. A few microbial





taxa show potentially PA and FL dual lifestyle strategies, able to switch according to substrate
availability an environmental variation and implying versatile metabolic flexibility. In addition,
according to some special microbial lineages supposed to be restricted in upper euphotic zones, we
found that the sinking organic particles likely function as vectors to transfer prokaryotes from surficial
ocean to deep waters, indicative of the potential vertical connectivity of prokaryotes along water
column profile.

## Data availability

The pyrosequencing data obtained from the 454 sequencing of 16S rRNA genes were deposited in the
Sequence Read Archive (SRA) database under accession ID PRJNA546072 for bacterial sequences
and PRJNA546071 for archaeal sequences.

## Author contribution

JL and JF designed the experiments, and JL, LG, JW and BW carried them out. JL, SB, LZ and LS
treated and analyzed the sequence data. JL and JF wrote the manuscript with contributions from all co-
authors.

## Acknowledgements

This work was financially supported by the National Natural Science Foundation of China (NSFC, No.
41373071 and No. 91951210) and National Key R&D Program of China (No. 2018YFC0310600).

## Competing interests

The authors declare that they have no conflict of interest.



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



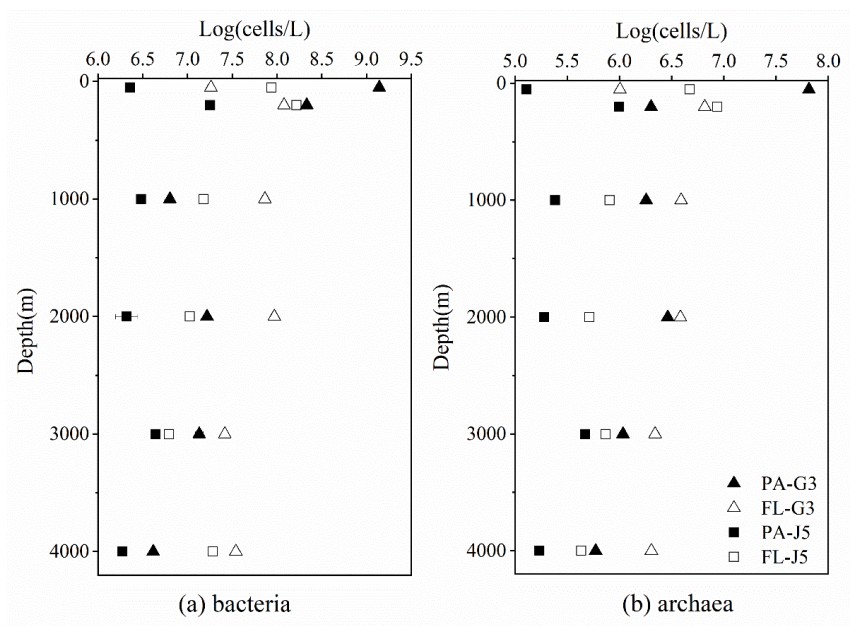



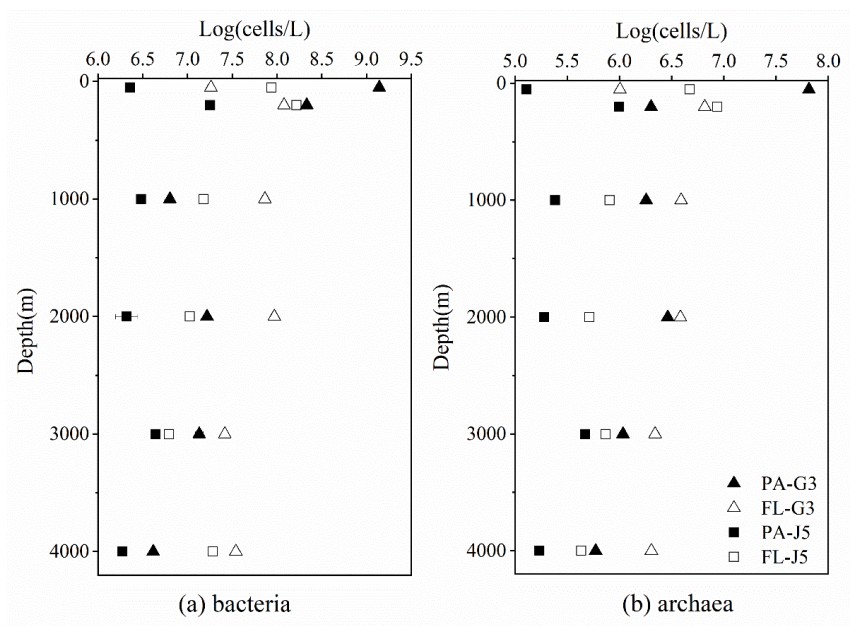


**Figure 1.** Bacterial and archaeal cell abundances in seawaters at different depths from G3
station and J5 station in the South China Sea, estimated from 16S rRNA gene copy
abundances.






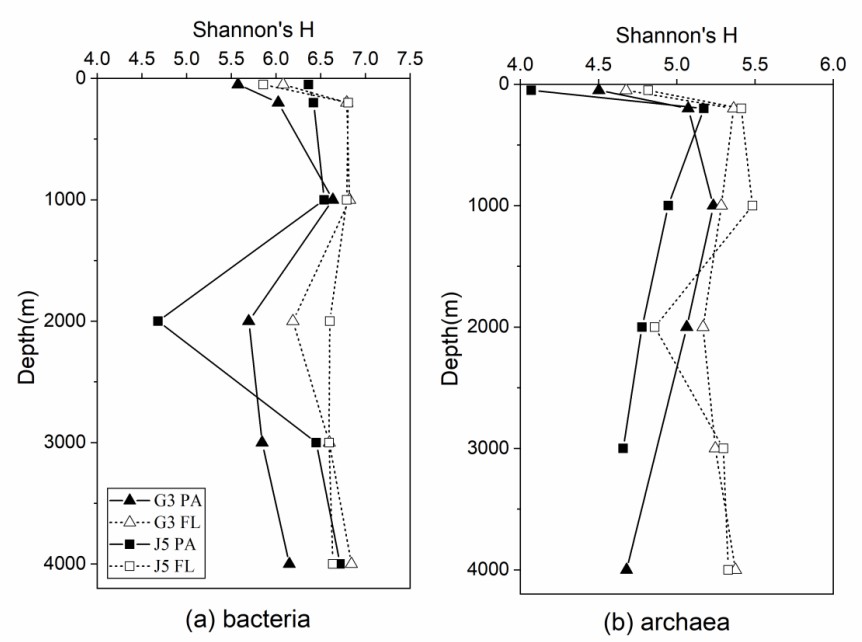


Figure 2. Shannon's diversity index calculated for all bacterial and archaeal communities of
seawaters collected from G3 station and J5 station in the South China Sea.



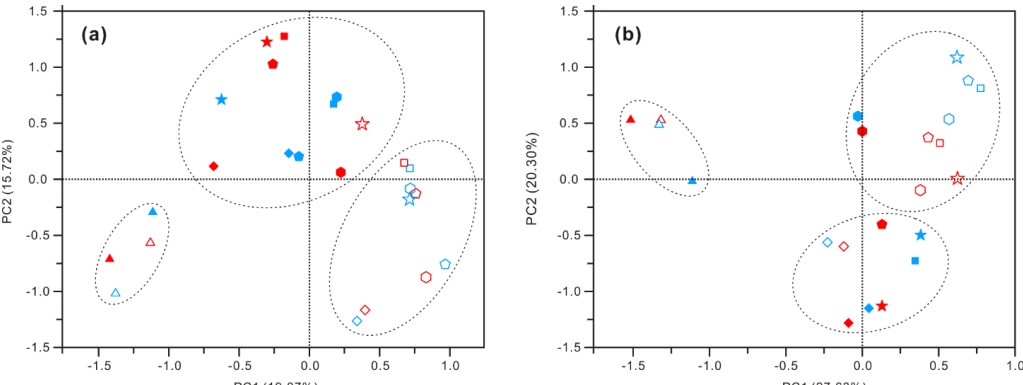

**Figure 3.** Results of PCoA analysis for particle-attached and free-living microbial fractions collected from seawater columns of the South China Sea. (a) PA and FL bacteria; (b) PA and FL archaea. Triangle: 50 m; rhombus: 200 m; hexagon: 1000 m; star: 2000 m; square: 3000 m; pentagon: 4000 m. Blue color: J5 station; red color: G3 station. Filled: particle-attached fraction; open: free-living fraction.



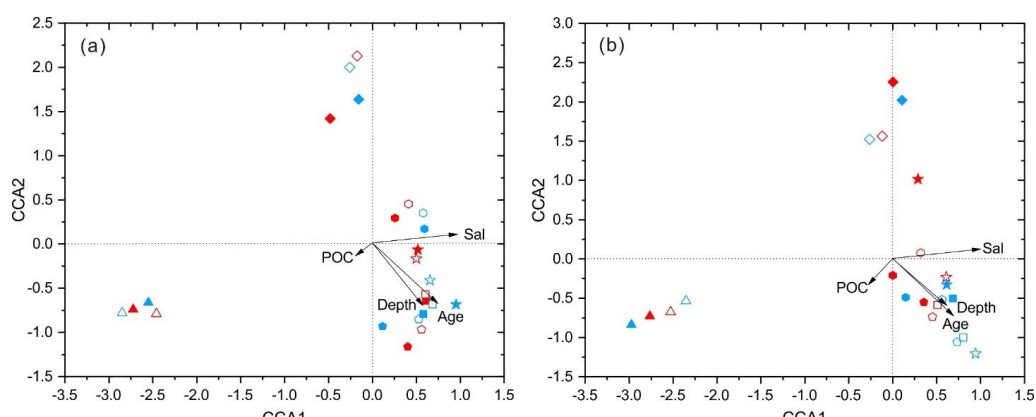

**Figure 4.** Results of CCA analysis to correlate several environmental factors including POC, seawater age,
salinity and depth to PA and FL microbial communities collected from seawater columns of the South China
Sea. (a) PA and FL bacteria; (b) PA and FL archaea. Triangle: 50 m; rhombus: 200 m; hexagon: 1000 m; star:
2000 m; square: 3000 m; pentagon: 4000 m. Blue color: J5 station; red color: G3 station. Filled: particle-
attached fraction; open: free-living fraction.
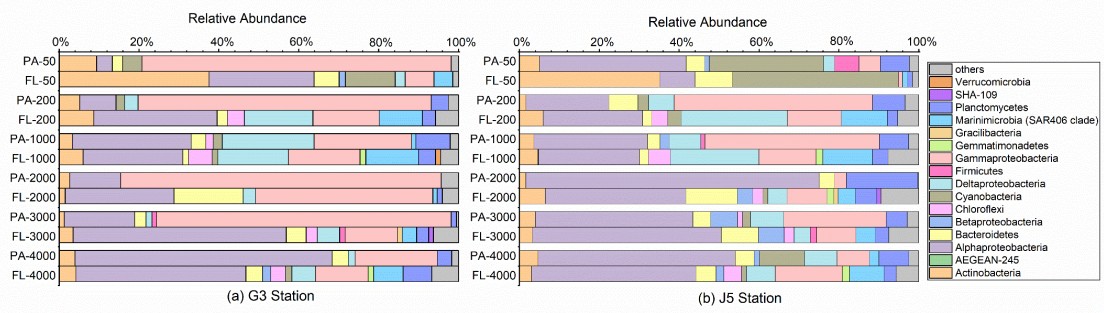

**Figure 5.** Taxonomic compositions of particle-attached and free-living bacterial communities of seawaters at different depths along two different water columns in the South China Sea. (a) G3 station; (b) J5 station. The phylum or class which has less than 1% proportions is classified into "others" (Fig. S4).

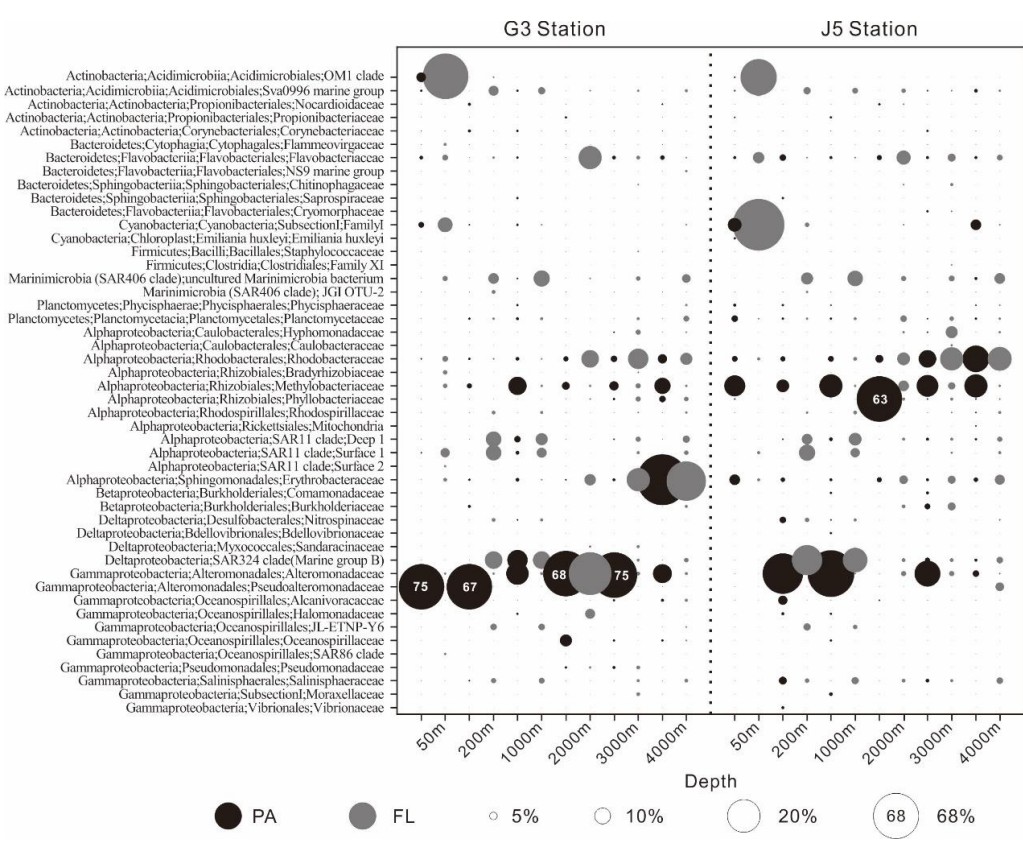

**Figure 6.** The relative abundances of families in particle-attached and free-living bacterial communities. Dark grey bubbles are the average relative abundances in the PA fraction, while light grey bubbles are the average relative abundances in the FL fractions. Scale is shown in the bottom, and the cycle with a number inside indicates actual relative abundance.



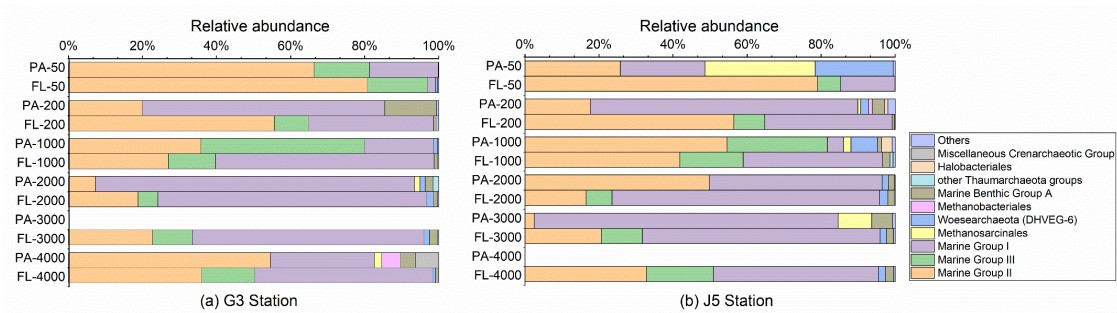


**Figure 7.** Taxonomic compositions of particle-attached and free-living archaeal communities of seawaters at


different depths along two different water columns in the South China Sea. (a) G3 station; (b) J5 station. The


archaeal lineages, at ~ phylum or class level, with less than 1% proportions is classified into "others" (Fig. S5).


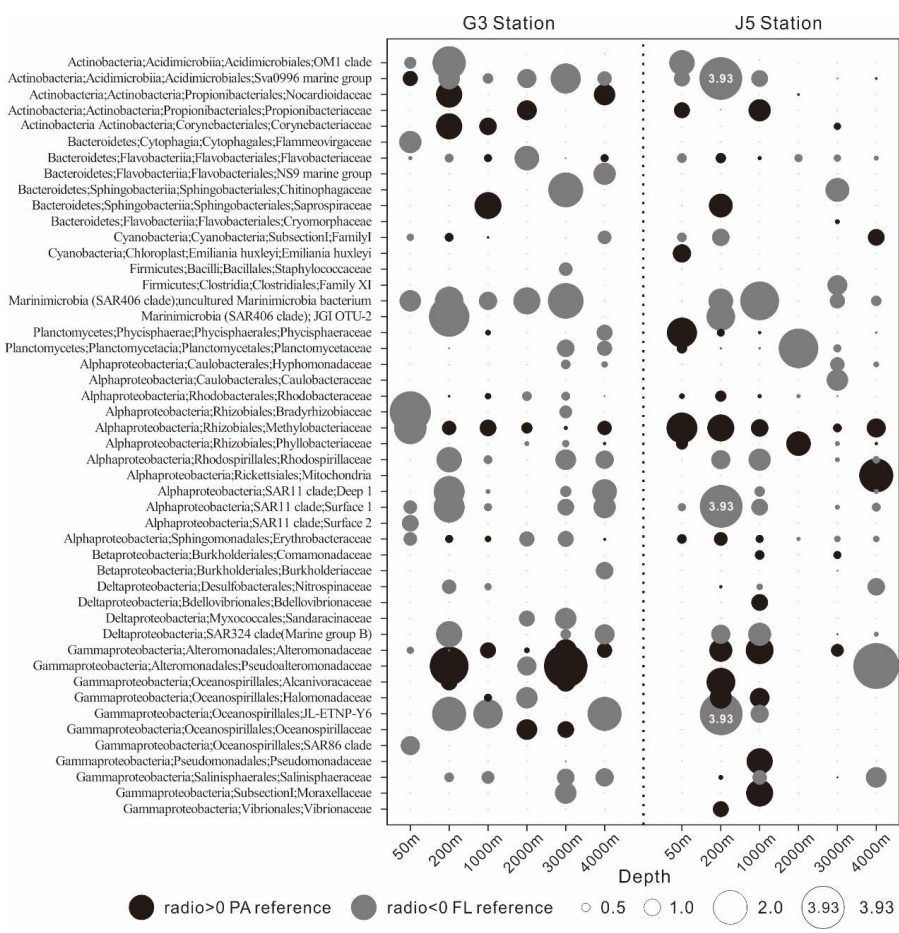

**Figure 8.** Odds ratio for each of the families with relatively abundant proportions in each sample.
Dark grey bubbles represent clades with a positive odds ratio, meaning higher relative abundance in
the PA fraction. Light grey bubbles represent clades with a negative odds ratio, or higher relative
abundance in the FL fraction. Scale is shown in the bottom, and the circle with a number inside
indicates actual ratio (not proportional).

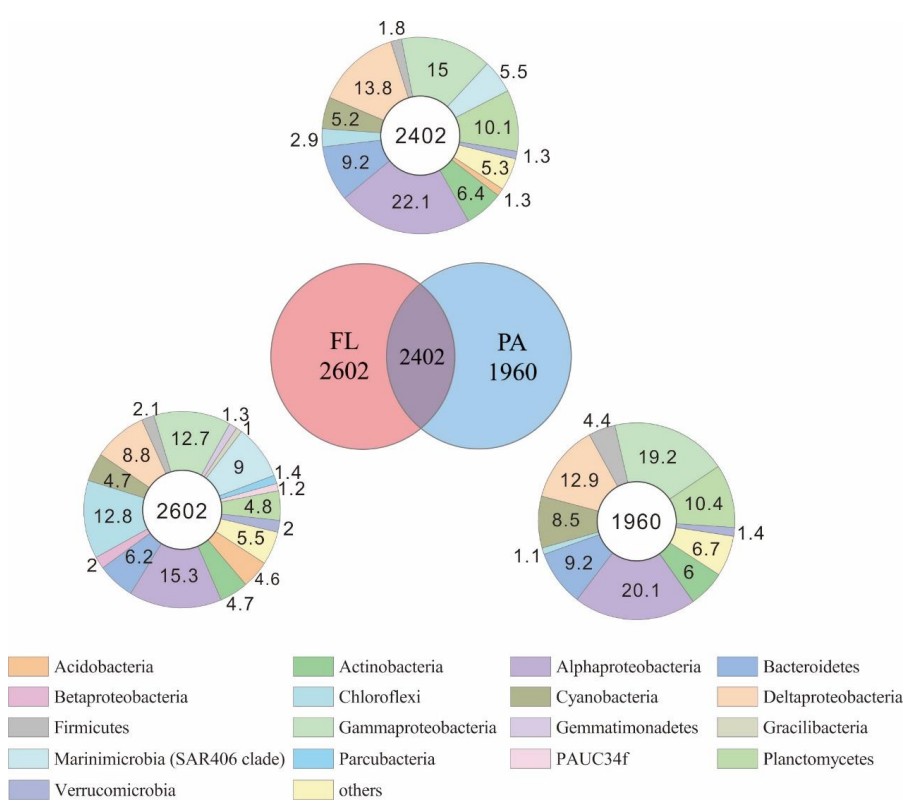

**Figure 9.** Numbers of each OTU sets including those exclusively found in PA fraction, FL fraction, and those shared by PA and FL fractions. Pie charts represent relative proportions of each bacterial lineages at phylum/class level.





**Table 1.** Environmental parameters of the water columns at different depths of G3 and J5 stations in the South China Sea

| Depth (m) | G3 station | | | | | | | | | | J5 station | | | | | | | | | |
|---|---|---|---|---|---|---|---|---|---|---|---|---|---|---|---|---|---|---|---|---|
| | T (°C) | Sal. (‰) | pH | DO (uM) | DOC (µM) | POC (µM) | Ages* (yr) | $NO_3^-$ (µM) | $PO_4^{2-}$ (µM) | Silicates (µM) | T (°C) | Sal. (‰) | pH | DO (uM) | DOC (µM) | POC (µM) | Ages* (yr) | $NO_3^-$ (µM) | $PO_4^{2-}$ (µM) | Silicates (µM) |
| 50 | 25.80 | 33.81 | 8.02 | 204.3 | 63.07 | 1.5 | 109 | BD | BD | 2.27 | 23.60 | 33.88 | 8.02 | 204.8 | 67.77 | 1.6 | 108 | 0.12 | BD | 2.36 |
| 200 | 15.46 | 34.54 | 7.75 | 115.1 | 53.02 | 0.8 | 106 | 17.98 | 1.20 | 21.06 | 14.27 | 34.52 | 7.72 | 116 | 49.99 | 0.9 | 106 | 19.13 | 1.30 | 26.56 |
| 1000 | 4.68 | 34.51 | 7.51 | 85.5 | 49.34 | 1.2 | 1170 | 37.16 | 2.72 | 114.40 | 4.46 | 34.53 | 7.51 | 82.3 | 45.62 | 2.1 | 1310 | 37.04 | 2.73 | 121.93 |
| 2000 | 2.52 | 34.61 | - | - | - | 1.1 | 1190 | - | - | - | 2.49 | 34.61 | 7.52 | 102 | 41.67 | 0.9 | 1670 | 38.41 | 2.81 | 151.46 |
| 3000 | 2.36 | 34.62 | - | - | 42.94 | 1.8 | 1600 | - | - | - | 2.36 | 34.62 | 7.52 | 109.7 | 40.34 | 0.7 | 1680 | 38.16 | 2.79 | 145.03 |
| 4000 | 2.39 | 34.63 | 7.52 | 115.1 | 42.44 | 0.7 | 1750 | 38.48 | 2.82 | 141.81 | 2.43 | 34.62 | 7.53 | 111.8 | 46.52 | 1.2 | 1610 | 38.58 | 2.78 | 145.06 |

*$\Delta^{14}C$ ages; BD: Below detection; -: no measurement.