# Peer review of "Characterization of particle-associated and free-living bacterial and"

_Biogeosciences, 2020_

## Referee Comment (RC1) · Anonymous Referee #1 · 11 May 2020

Li et al investigated particle-attached (PA) and free-living (FL) bacterial and archaeal community structures in South China Sea. They quantified the abundance of bacteria and archaea by using qPCR and surveyed the community structure with pyrosequencing. High abundance and diversity of FL than PA were observed. They tried to related microbial community composition, life styles and environmental adaption to organic and inorganic substrate availability from surface to deep ocean. Major concern: The present MS is a little bit "microbial", not "biogeochemical". It will be great to include organic chemical analysis of particles and waters if any. At least, discuss this based on data available in previous studies. I suggest to discuss possible technique bias including 1) filtration with 3 um to collect particles, especially for deep sea samples which is

very fragile. 2) qPCR data which showed relatively low "cell abundance" compared to microscopy. Having age data of particles is very interesting. I encourage the authors discuss more about this and its relationship, and biogeochemical implicates, with microbial data. A logic is needed to explain the sinking rate and age of particles as well as microbes attached.

Specific comments: Sometimes the "recently" is not appropriate since the references are not recent at all (e.g. Line 59, Line 460). Provide methods for particle age measurement. Salinity does not have unit (e.g. Line 200). Include statistical analysis (e.g. Line 219). Line 240, seems meaningless to point out the number of sequences per depth. Line 365, any evidence or previous study to support the different origins of organic matter of G3 and J5? Line 404, I understand that POM remineralization is oxygen dependent, but the cause and effect relationship between DO concentration and particle flux is not clear to me. Line 462: li? Maybe use copy number, not cell abundance, throughout the MS.

---

## Author Comment (AC1) · 19 Jun 2020

We have revised our manuscript according to the managing editor's comments generated in the early phase of the Discussion time. Specifically, we modified parts of the M&M section (i.e., protocols for DNA extraction and real-time PCR) where some of the sentences appear to be similar to our early papers. Additionally, we provided more protocol details in the revised version of the manuscript. Please use the revised version for further discussion and commenting.

Please also note the supplement to this comment:

https://www.biogeosciences-discuss.net/bg-2020-115/bg-2020-115-AC1-supplement.pdf

[Figure]

**Supplement:**

[revised manuscript text omitted]

---

## Referee Comment (RC2) · Anonymous Referee #2 · 7 Jul 2020

General comments

This manuscript by Li et al. is an examination of the PA and FL microbial communities found throughout depth at two stations in the South China Sea, and is an interesting addition to the body of literature on particle association of ocean microbes. The general patterns found in the microbial community composition data are reasonable. However, it is unclear whether the authors performed specific important transformations of the count table data before statistical analyses. Without that point being clarified, I would be very cautious to interpret anything from the ordinations and diversity calculations. The authors also included an analysis of seawater age, which is a unique aspect of

this dataset. I would like to see a bit more exploration of that in relation to specific microbial taxa. Generally, I think this is an interesting and publishable dataset, but some refinement of the statistical methods are necessary.

Specific Comments

L107- A little context on the stations would be nice. They seem to be part of a larger study. What is their significance and why were these two chosen?

L172 This is a very outdated version of SILVA. I'm not going to argue that the classification should be redone, but there are likely implications that can be discussed (eg. it may explain the large amount of unidentified archaeal taxa. Also, another example: the Nitrospinaceae are no longer considered part of the delta-proteobacteria, but in their own Nitrospinae phylum, L348).

Section 2.4- There is no mention of transformation/normalization of count tables or removing singletons. Removing singletons is absolutely vital for analyzing OTU data because 97% clustering introduces lots of singleton artifacts (Edgar RC. 2017. Accuracy of microbial community diversity estimated by closed- and open-reference OTUs. PeerJ 5:e3889. DOI: 10.7717/peerj.3889), and this could greatly skew estimates of diversity and ordination results. Removal of singletons will also change the results for Figure 9 and the diversity estimates. Transformation is absolutely necessary for ordinations (see Legendre and Gallagher. 2001. Ecologically meaningful transformations for ordination of species data. Oecologia 129:271–280. DOI: 10.1007/s004420100716 and Gloor GB, Macklaim JM, Pawlowsky-Glahn V, Egozcue JJ. 2017. Microbiome Datasets Are Compositional: And This Is Not Optional. Front Microbiol. 8(NOV):1–6. doi:10.3389/fmicb.2017.02224), so it needs to be made clear if this was done or not. Tables S1 and S2 appear to be raw count data with no transformation or normalization

Section 2.5- No quality parameters of the qPCR assays are reported (eg. $R^2$ of the standard curve or efficiency of the reaction). Also what standard was used for qPCR? A PCR product? Genomic DNA from cultured organism with a known 16S rDNA copy

number? This should be briefly mentioned.

L249-256 & L377 It's very interesting that diversity decreased mid-water column and then increased again below that. Can the authors speculate what's going on here? Could they relate it to their DOC/ POC data or age of seawater?

L257- 259 & Fig. 3 I see the separation of the 3 identified groups in the ordination but it is unclear which test was used to statistically distinguish these groups or if the circles were just drawn based on looking at the figure.

L410-432 Since the authors analyzed the age of seawater, it would be nice to interpret this more directly with respect to DOC/POC quality and microbial community composition. What is the precise impact on microbial community composition based on age of seawater (which groups were important and why?). I like that this part of the discussion begins to interpret the impact of silicate (which is really an indirect correlate and likely a signal of diatom biomass impacting microbial community, as the authors begin to suggest). But I think this can go deeper given the high-resolution community composition data that is available here (similar to the detailed discussion on PA/ FL preference).

Figure 9 is not introduced in the results but heavily discussed in the discussion. The results reported for Fig. 9 in the Discussion should be moved to the Results.

L602- Bchl a is introduced for the first time with no context on what this is or what it is short for.

Technical Corrections

L37- A high proportion "of" overlap

L140- What is CTAB?

L151- "each DNA was" should be each "DNA pellet was"?

L259- I am not sure what is meant by incompact.

L388: "were supposed to" is a misleading phase. It sounds like an expectation of a result. Perhaps this would better be "several environmental parameters played a pivotal role..."

L403 impaction should be impact

L412, "It is considered..." I am not sure what the 'subset' is and I think this can be better phrased.

L414-415- should be 'A recent study' (not 'A most recent study')

L425 – should be 'unexpected' rather than 'out of our expectation'

L425-426 – should be 'generally exhibits N- or P-limited phytoplankton production'

L436- 'niches' is not the correct word here. Maybe habitats? Locations?

L445- The phrase 'significantly divergent' implies statistical significance, but no such test was done to prove that PA and FL communities were significantly different (also in lines 641, 27, and 103). I think just 'divergent' would be acceptable unless a test is incorporated.

L463- 'dominantly govern' should just be 'dominate'

L498 – I don't understand the meaning of this phrase: 'nothing is available to elaborate the selection better PA and...' I think it needs to be reworded.

L580- The phrase 'intelligibly convinced' is unclear. Also the entire sentence L580-583 is a run-on sentence with some unclear phrasing and I'm not sure what the intended meaning is.

---

## Referee Comment (RC3) · Anonymous Referee #3 · 21 Jul 2020

This study focused on the depth profiles of free-living (FL) and particle-attached (PA) prokaryotes (Bacteria and Archaea) in two sites in the South China Sea (SCS). As of now, there is a few studies to reveal the particulate-attached prokaryotic community structures (especially, about Archaea). 16S rRNA gene deep-sequencing analyses revealed the shift of bacterial and archaeal community structures among different depths. Also, several environmental factors such as depth, seawater age, salinity, POC, DOC, DO, and silicate could be critical for determining the community structures. Phylogenetic analyses revealed that several lineages including alpha-, gamma-proteobacteria, Actinobacteria, Bacteroidetes . . .etc. were overlapped between PA and FL fractions. However, there were differences at family level among them. According

to these data, the authors discussed about ecological and biogeochemical roles of FL and PA prokaryotes in the SCS.

Major comments The manuscript is well written. But main limitation is a weak of biogeo-scientific discussion of FL and PA (especially PA) fractions (e.g. interaction between chemical composition or degradation of POM, and PA bacteria or archaea). In addition, the critical problem is potential primer biases (especially bacterial primer). The selection of primer set is very important for evaluating the prokaryotic community structure and diversity. Especially, SAR11 clade affiliated with Alphaprotoebacteria seems to be underestimated in this study. This clade is known to be dominant lineage in the oceanic environments, and generally accounted for 15∼30% of total prokaryotic cells (Morris et al., 2002 Nature 420: p806-810). Different primer set create different results (e.g. Sanchez et al., 2009, Aquat Microb Ecol, 54: p211-216; Apprill et al., 2015, Aquat Microb Ecol, 75: p129-137) on the community analysis in the ocean (at least Bacterial community analysis). The authors should mention these problems in the discussion section.

Provide more information on the choice of sites and depths for this work. Moreover, provide more detail profiles of environmental factors collected by a Sea-Bird CTD system (at least seawater temperature, salinity and DO).

L152-155: Why did the authors choose these primer set (especially, 27F-533R for Bacteria)? I think the SAR11 clade affiliated with Alphaproteobacteria were underestimated (approximately 15∼30% of total 16S rRNA sequences, in general). The selection of primer is one of the most critical factors for evaluating prokaryotic community structures and diversity.

Provide the data for sequence depth (e.g. rarefaction curves) of 16S rRNA gene used in this study.

Specific comments L96: Marine Group III (MGII)→Marine Group (MGIII) L150: what amount of template DNA (ng) did author used? And provide the information about DNA

concentration (or amount) after DNA extraction. L152-155: provide the references of these primers. L161: provide the reference or URL for QIIME 1.9.1 software L177: provide the reference or URL for R packages. L251 and other lines: If the author described "significantly differences", provide the information about R or Rho values, and P value. Maybe, the ANOSIM or PERMANOVA analyses should be need to clarify statistical differences among communities. L379 "taxonomically" : add information about the taxonomic levels after this word (e.g. taxonomically (at least family or order?? level)). L386 "depth": I think it is better to correct "hydrological condition (e.g. depth)". L413: provide the R or Rho value before P value (R or Rho=????, P>0.05). L417: I can not understand "utilization of refractory POC by microorganisms depends on the quality of POC". I recognize "refractory" is not usable for microorganisms. "Refractory POC" means "POC in the deep sea"? L442-443: $\gamma$-proteobacterial (change italic to regular) L446-449: Again, primer selection is one of the critical factors for evaluating the community composition and diversity. Thus, the authors should add the discussion about primer biases. L462: What is (li)? Reference? L522-523 "statistical analysis": provide the R or Rho, and P values.

Figure 3: the authors circled the points (triangles, rhombus+hexagon+star...etc.) for representing different clusters in bacterial and archaeal fractions. Are there statistically significant? Provide the results of statistical analyses (and show R or Rho value, and P value). Figure 6: x-axis is confused. It is better to delete minor scale marks (e.g. those between 50m and 200m). Figure 7, Figure S5: provide the information about failed samples in the legend.

---

## Author Comment (AC2) · 9 Sep 2020

Anonymous Referee #1 Li et al investigated particle-attached (PA) and free-living (FL) bacterial and archaeal community structures in South China Sea. They quantified the abundance of bacteria and archaea by using qPCR and surveyed the community structure with pyrosequencing. High abundance and diversity of FL than PA were observed. They tried to related microbial community composition, life styles and environmental adaption to organic and inorganic substrate availability from surface to deep ocean.

Major concern: The present MS is a little bit "microbial", not "biogeochemical". It will be great to include organic chemical analysis of particles and waters if any. At least,

discuss this based on data available in previous studies (1). I suggest to discuss possible technique bias including 1) filtration with 3 um to collect particles, especially for deep sea samples which is very fragile (2). 2) qPCR data which showed relatively low "cell abundance" compared to microscopy (3). Having age data of particles is very interesting. I encourage the authors discuss more about this and its relationship, and biogeochemical implicates, with microbial data. A logic is needed to explain the sinking rate and age of particles as well as microbes attached (3).

Response to comment (1): Thanks for this suggestion.We agree in that our manuscript is mainly focused on the microbiological part and the role of microbes in marine carbon cycle. On the "biogeochemical" part, we focus our discussion on the role of PAM and FLM in oceanic carbon cycling processes, i.e., decomposition of POM, inter-conversion between POM and DOM, and degradation of DOM. To this end, the present study is an extension of our previous work, focusing on both microbiological and biogeochemical aspects of PAM and FLM and their potentials in mediating carbon cycling processes in the ocean. Therefore, revealing the microbial taxa in PA and FL assemblages and profiling variations of their abundance and diversity along the water column provides a foundation for a better understanding of the coupled microbiological and biogeochemical processes in marine carbon cycle. In the manuscript, we had/added additional discussion on microbial metabolic potential in utilizing certain organic compounds. For examples: "They often maintain ......, and are capable of degrading high-molecular-weight (HMW) organic compounds......." "It is further revealed that PA microbes ...... metabolic and regulatory capabilities of utilizing compositionally varied organic matter, while ......" "These $\gamma$-proteobacterial members are ...... they are believed to have the abilities to degrade/utilize HMW organic compounds with higher nutrient requirements." "Further phylogenetic assignment revealed ...... belong to the genus Methylobacterium which are strictly aerobic, facultatively methylotrophic bacteria, and can grow on a wide range of carbon compounds." "Genomic information underlines that although these clades have a flexible metabolism utilizing multiple hydrocarbon compounds......" "The majority of ......, and commonly possess the ability

to hydrolyze and utilize complex carbon sources. Although their abundance . . . . . . because of their high specificity for organics." "Sva0996 marine group . . . . . . have the ability to assimilate phytoplankton-derived dissolved protein." . . . . . . . . . . . .

Response to comment (2): Yes, the reviewer is right and we agree. About the criteria to distinguish the PA and FL microbial assemblages, there are different standards about pore-size of filtering membrane such as 3 $\mu$m, 1$\mu$m and 0.8/0.7 $\mu$m. By now, the 3.0 $\mu$m nominal pore size is most commonly used. In addition, as pointed out by the reviewer, particles including organic detritus and meiofauna such as metazoans and protists seem to be very fragile and precarious (Lecroq et al., 2011; Bochdansky et al., 2017) and it is inevitable to break them if the filtering process is intensive. Therefore, to avoid damaging fragile particles (and membrane) in our experiment, we use a relatively low vacuum pressure of < 10 mm Hg, and at the same time, the filtration time was less than 40 minutes, which has been confirmed as an effective way. In the M&M section of our manuscript, we added one more sentence to provide this method detail: "To avoid damaging the membrane and fragile particles, a relatively low vacuum pressure of < 10 mm Hg was used, and at the same time, the filtration time was less than 40 minutes."

Response to comment (3): We respectfully disagree. The microbial abundances estimated by qPCR of 16S rRNA gene approximately equal to the results of staining under microscope (for example, see the results in Zhang et al., 2020, Marine Pollution Bulletin). As we described in M&M section and our response to the last comment (below), although there are some biases in converting 16S rRNA gene copy numbers into bacterial and archaeal cell abundances which mainly results from the significantly different copy numbers of 16S rRNA gene in different taxa, the estimation of cell abundances based on qPCR results of 16S rRNA gene can reflect the approximate biomass of cell abundances and have been widely used. During our sampling, because we did not fix the samples with PFA, so, we used the qPCR to roughly estimate the cell abundances of different size fractions. However, as suggested by the reviewer, we added a few

sentences to point out this potential biases of this kind of estimation: "Although the cell abundances inferred from the 16S rRNA gene copy number quantified by qPCR may be potentially biased, the estimation of cell abundances based on the qPCR of 16S rRNA gene has been confirmed as an effective method to reflect the approximate cell abundances in previous studies."

Response to comment (4): Thanks for this advice. However, we think the reviewer misunderstood our dataset. The age dataset in our manuscript is the ages of seawater at different depths rather than ages of organic particles. The age of seawater was determined based on the radiocarbon dating of DIC instead of organic carbon from particles.

Specific comments: Sometimes the "recently" is not appropriate since the references are not recent at all (e.g. Line 59, Line 460). Our response: Yes, we agree. We have corrected these points by deleting "recently".

Provide methods for particle age measurement. Our response: Thanks for pointing out this. The dating was performed in Beta Analytic (Miami, United States). We provided this method in our M&M section as below: "1 L of seawater for each sample were sent to Beta Analytic, Inc. in Miami, Florida, for 14C radiocarbon dating with the Accelerator Mass Spectrometry (AMS) methods as described in their website (https://www.radiocarbon.com/beta-lab.htm). When CTD rosette sampler came back on board, seawater for 14C dating was taken from Niskin bottles with first priority. During the sampling, to avoid the disturbance of air, glass bottles were fully filled with seawater with no headspace. In addition, mercury chloride was added to prevent any microbiological influence."

Salinity does not have unit (e.g. Line 200). Our response: Thanks. We added the "PSU" as the salinity unit.

Include statistical analysis (e.g. Line 219). Our response: Thanks for pointing out this. In our Fig. 1, the standard deviations (SD) were actually provided, but because

most SD values are too small that they are not shown up clearly on the graph. In our maintext, we provided these related information of SD in the subsection of "3.2 Microbial cell abundances".

Line 240, seems meaningless to point out the number of sequences per depth. Our response: We agree, and therefore, we deleted these numbers of bacterial and archaeal sequences.

Line 365, any evidence or previous study to support the different origins of organic matter of G3 and J5? Our response: As we stated in the manuscript, it is our hypothesis to speculate the potential influence from POC quality, but without direct evidence in our study. Geographically, G3 site was close to the northern South China Sea, i.e., near the continent, while J5 was in the southern South China Sea, although they both were located in the central basin of the SCS. It has been shown that the Pearl River plume could reach nearby area of the G3 site (He et al., 2016), and moreover, there are more eddy activities around the northern SCS basin (Xiu et al., 2010). Additional allochthonous nutrient inputs from river discharge and eddy pumping could bring multifarious organic particles with different compositional characteristics. In addition, the enhancement of additional nutrient supplies can further irritate the growth (even the blooming) of phytoplanktons at G3 station and shape their community compositions which dominate the organic composition (quality) of POM in seawaters, especially in the surface water. Several researches have revealed significant differences in phytoplankton size structure (Chen et al., 2015; Lian et al., 2018) and community composition (Ke et al., 2009, 2012) between the southern and northern South China Sea. All these indicate a possibility that there may be some differences in the quality of POC between G3 and J5 sites. Therefore, we cited a couple of these references to support our hypothesis.

Line 404, I understand that POM remineralization is oxygen dependent, but the cause and effect relationship between DO concentration and particle flux is not clear to me. Our response: As shown in a couple of studies, DO is an important environmental

variable that impacts organic particle flux by affecting respiration rates of particle-associated microbes (Kalvelage et al., 2015), and thus, the remineralization rate of organic particles and transfer efficiency and flux of sinking POM (Marsay et al., 2015; Cram et al., 2018).

Line 462: li? Our response: It is a typo and here should be a reference, Gong et al., 1992. We had corrected this mistake.

Maybe use copy number, not cell abundance, throughout the MS. Our response: Thanks for this advice but we respectfully disagree. To provide a direct comparison of cell abundances, we converted the copy number of 16S rRNA gene into cell abundance based on the average values of 16S rRNA gene copy number in bacteria and archaea. In such case, it is relatively easy and intuitive to compare the abundances of bacteria and archaea among different size fractions. Therefore, we keep this conversion about cell abundances.

Please also note the supplement to this comment:
https://bg.copernicus.org/preprints/bg-2020-115/bg-2020-115-AC2-supplement.zip

———————————————————

[Figure]

**Fig. 1.** Bacterial and archaeal cell abundances in seawaters at different depths from G3 station and J5 station in the South China Sea, estimated from 16S rRNA gene copy abundances.

**Fig. 2.** Shannon's diversity index calculated for all bacterial and archaeal communities of sea-waters collected from G3 station and J5 station in the South China Sea.

(a) bacteria

(b) archaea

[Figure]

**Fig. 3.** Results of PCoA analysis for particle-attached and free-living microbial fractions collected from seawater columns of the South China Sea. (a) PA and FL bacteria; (b) PA and FL archaea.

[Figure]

**Fig. 4.** Results of CCA analysis to correlate several environmental factors including POC, seawater age, salinity and depth to PA and FL microbial communities collected from seawater columns of the SCS.

[Figure]

**Fig. 5.** Taxonomic compositions of particle-attached and free-living bacterial communities of seawaters at different depths along two different water columns in the South China Sea. (a) G3 ; (b) J5.

[Figure]

**Fig. 6.** The relative abundances of families in PA and FL bacterial communities. Dark grey bubbles are for the PA fraction, while light grey bubbles are for the FL fraction.

[Figure]

**Fig. 7.** Taxonomic compositions of particle-attached and free-living archaeal communities of seawaters at different depths along two different water columns in the South China Sea. (a) G3 ; (b) J5.

[Figure]

**Fig. 8.** Odds ratio for each of the families with relatively abundant proportions in each sample.

---

## Author Comment (AC3) · 9 Sep 2020

General comments This manuscript by Li et al. is an examination of the PA and FL microbial communities found throughout depth at two stations in the South China Sea, and is an interesting addition to the body of literature on particle association of ocean microbes. The general patterns found in the microbial community composition data are reasonable. However, it is unclear whether the authors performed specific important transformations of the count table data before statistical analyses. Without that point being clarified, I would be very cautious to interpret anything from the ordinations and
diversity calculations (1). The authors also included an analysis of seawater age, which is a unique aspect of this dataset. I would like to see a bit more exploration of that in relation to specific microbial taxa (2). Generally, I think this is an interesting and publishable dataset, but some refinement of the statistical methods are necessary (3).

Response to comment (1): Thanks for pointing out this and we agree. In our original manuscript, we did not describe the details about our statistical analyses such as PCoA and CCA in the M&M section. During our this revision, basic information about these methods were provided. In brief, we first removed all the singletons from our OTU tables. Then, to avoid the variation caused by an unequal sequence number across samples, we normalized the OTUs abundance by resampling the sequences for each sample based on the sample with the least number of sequences. After resampling the sequences to the same number, alpha diversity including Chao 1 and Shannon was calculated and then used to compare diversity between different samples. For the $\beta$-diversity such as PCoA and CCA ordinations, we performed the transformation of the resampled OTU abundance by taking the log of the sequence numbers. All the details about these analyses were provided in our revised M&M section: "To avoid the variation caused by an unequal sequence number across samples, the OTUs abundance was normalized by resampling sequences for each sample based on the sample with the least number of sequences. After resampling the sequences to the same number, diversity estimators including Chao 1 and Shannon's diversity (H) were calculated. Similarities among different microbial communities were determined using similarity matrices generated according to the phylogenetic distance between reads (Unifrac distance), and beta diversity of principal coordinates analysis (PCoA) was computed as components of the QIIME pipeline. The correlation between the microbial community structures and environmental parameters was analyzed by canonical correspondence analysis (CCA). For the PCoA and CCA ordinations, the transformation of the resampled OTU abundance table was performed by taking the log of the sequence numbers. In addition, to test the statistical significance of different groups identified by PCoA ordination, multiple statistical analyses including MRPP, ANOSIM and PERMANOVA were

performed based on the resampled and transformed OTU abundance table. Mantel test was also performed to test the statistical significance of environmental factors with microbial community compositions from the results of CCA. All statistical analyses were performed in the R environment (v 3.2.1) using the Vegan package (https://CRAN.R-project.org/package=vegan)."

Response to comment (2): Agree and done. Please see our response to the comment below about line 410-432.

Response to comment (3): Thanks for the informative comments. As we responded above, we reanalyzed our data and also provided the detailed information of statistical analyses.

Specific Comments L107- A little context on the stations would be nice. They seem to be part of a larger study. What is their significance and why were these two chosen?

Our response: The present work is motivated by our early works (Li et al., 2015) in which some preliminary findings indicated that depth probably exert an impact in structuring microbial assemblages. Therefore, in our present research, we selected two stations in the central basin of the SCS with depths >4,000 m to take the samples and test our hypothesis. The following sentence was added in "2.1 Sample collection and environmental parameter measurements" subsection to introduce this background: "Both stations have depth > 4,000 m, providing us the bathyal environments to vertically profile the variation of microbial assemblages with depth."

L172 This is a very outdated version of SILVA. I'm not going to argue that the classification should be redone, but there are likely implications that can be discussed (eg. It may explain the large amount of unidentified archaeal taxa. Also, another example: the Nitrospinaceae are no longer considered part of the delta-proteobacteria, but in their own Nitrospinae phylum, L348).

Our response: Thanks for pointing out this. The version of SILVA database used for

our study was actually 128 rather than 119 for the annotation of 16S rRNA gene sequences. However, even in the 128 version, the family Nitrospinaceae is still assigned into the class $\delta$-Proteobacteria which is, as said by the reviewer, outdated. Therefore, during our revision, we have reanalyzed all our OTUs based on the latest 132 version of SILVA database. Only a few variations occurred in bacterial and archaeal community compositions at $\sim$ phylum or class levels compared with our original results (Fig. 5, Fig. 7 and supplementary Fig. S3 and S4). It should be pointed out that in the latest 132 version, we found some inconsistent annotations with known taxonomic classifications at $\sim$ family level. So, we double checked all the dominant lineages with manual curation.

Section 2.4- There is no mention of transformation/normalization of count tables or removing singletons. Removing singletons is absolutely vital for analyzing OTU data because 97% clustering introduces lots of singleton artifacts (Edgar RC. 2017. Accuracy of microbial community diversity estimated by closed- and open-reference OTUs. Peer J 5:e3889. DOI: 10.7717/peerj.3889), and this could greatly skew estimates of diversity and ordination results. Removal of singletons will also change the results for Figure 9 and the diversity estimates. Transformation is absolutely necessary for ordinations (see Legendre and Gallagher. 2001. Ecologically meaningful transformations for ordination of species data. Oecologia 129:271–280. DOI: 10.1007/s004420100716 and Gloor GB, Macklaim JM, Pawlowsky-Glahn V, Egozcue JJ. 2017. Microbiome Datasets Are Compositional: And This Is Not Optional. Front Microbiol. 8(NOV):1–6. doi:10.3389/fmicb.2017.02224), so it needs to be made clear if this was done or not. Tables S1 and S2 appear to be raw count data with no transformation or normalization.

Our response: Yes, we totally agree. Firstly, as suggested by the reviewer, we removed all the singletons from our OTUs tables (see supplementary Table S1 and S2) and then reannotated our OTUs based on the latest SILVA database as mentioned above. For the OTU tables, 1,982 singletons were removed from bacterial OTUs, and 329 singletons were deleted from archaeal OTUs. The sequences represented by

bacterial singletons only accounted for ∼ 0.2-1.4% of bacterial communities, and 0.07-0.3% of archaeal populations. Therefore, the removal of singletons did not affect our results of microbial community compositions (Fig. 5, 7 and Fig. S3, S4). Secondly, after the removal of singletons, we updated the results of statistical analyses such as PCoA and CCA ordinations and diversity estimation. As we responded to comment (1), for these statistical calculations, we performed the transformation or normalization of OTUs abundance tables. We resampled OTUs with sing_rarefaction.py for each sample to make all the samples have the same number of sequences. After resampling, alpha diversity including Chao 1 and Shannon was recalculated. For$\beta$-diversity such as PCoA and CCA ordinations, we also performed the transformation of the resampled OTU abundances by taking the log of the sequence numbers. All the details about these analyses were provided in our revised M&M section. Thirdly, supplementary Table S1 and S2 are provided with the original datasets of OTU information including names, abundances, annotating taxonomic classification at different levels, singletons and resampling results.

Section 2.5- No quality parameters of the qPCR assays are reported (eg. RËȨ2 of the standard curve or efficiency of the reaction). Also what standard was used for qPCR? A PCR product? Genomic DNA from cultured organism with a known 16S rDNA copy number? This should be briefly mentioned.

Our response: Thanks for pointing out these and we totally agree with the reviewer's opinion. The PCR products of bacterial and archaeal 16S rRNA gene were first cloned into a plasmid vector, and transformed into E. coli DH5a. The recombinant plasmids were extracted and purified. The obtained plasmid solution was adjusted to a concentration of about 100 ng/$\mu$L, and was subsequently diluted 10-folds with sterile water as the standards for qPCR reactions. Standard curves were acquired from 10-fold serial dilutions of standards. R2 for our qPCR amplifications varied between 0.994 and 0.996, indicating a strong linear relationship over the concentration ranges used in our study. The conversion between copy number of 16S rDNA and cell abundance is based

on the average values of known pure cultures of bacteria and archaea listed from the database as shown by Lee et al., 2009. As suggested by the reviewer, we mentioned all above information in our M&M section like below: "The PCR products of bacterial and archaeal 16S rRNA gene were first cloned into a pUC18 plasmid vector (Takara Bio Inc, Japan), and then transformed into E. coli. The recombinant plasmids were extracted and purified, and subsequently diluted 10-folds as the standards for real-time PCR reactions. R2 for the standard curves varied between 0.994 and 0.996, indicating a strong linear relationship over the concentration ranges used in our study."

L249-256 & L377 It's very interesting that diversity decreased mid-water column and then increased again below that. Can the authors speculate what's going on here? Could they relate it to their DOC/ POC data or age of seawater?

Our response: Thanks for this constructive comment. Yes, we also agree. It is an interesting observation that mid-water around 2000 m depth shows a lower diversity. One possibility is that 1500-2000 m is a rough boundary for different water masses in the deep, central basin of the South China Sea. Generally, the concentrations of POC and DOC gradually decreased with depth, causing a continuous decreasing in microbial diversity. However, the deep water mass (>2600 m) of the central basin comes from the western Pacific Ocean through Bashi Channel which is relatively rich in nutrients than the mid-water masses of SCS at shallow dapth. Therefore, it may cause a relative increase in microbial diversity in deep water masses such as those at 3000 m and 4000 m. In addition, some "old, deep" water from the bottom of the central basin will also rise to the 2000 m depth because of the basin-scale circulation. These old waters are relatively enriched in refractory DOC (RDOC), remained after microbial ultilization of labile OC during their cirlulation, potentially reducing microbial diversity. This hypothesis is supported by the seawater age at J5 station. It is shown that the age of seawater at 2000 m depth of J5 station is 1670 years, roughly equal to those of deep waters at 3000 m and 4000 m (1680 years and 1610 year).

L257- 259 & Fig. 3 I see the separation of the 3 identified groups in the ordination but

it is unclear which test was used to statistically distinguish these groups or if the circles were just drawn based on looking at the figure.

Our response: Yes. As pointed out by the reviewer, we identified different groups in our PCoA analysis based on the looking at the figure. To support the separation of these groups, we performed three more statistical analyses including MPPR, ANOSIM and PERMANOVA analyses. The results of these three analyses were listed in Table S3 of supplementary materials. They are statistically significant with P values <0.05. To clarify this statistical significance, we added this statistical support in the sentence as: "PCoA analysis revealed that there were significant differences (P <0.05, Table S3) in bacteria and archaea community structures over the depth profiles and between the FL and PA fractions." In addition, in the caption of Figure 3, one more sentence was also added: "Statistical analyses supported the grouping of the clusters (Table S3)."

L410-432 Since the authors analyzed the age of seawater, it would be nice to interpret this more directly with respect to DOC/POC quality and microbial community composition. What is the precise impact on microbial community composition based on age of seawater (which groups were important and why?). I like that this part of the discussion begins to interpret the impact of silicate (which is really an indirect correlate and likely a signal of diatom biomass impacting microbial community, as the authors begin to suggest). But I think this can go deeper given the high-resolution community composition data that is available here (similar to the detailed discussion on PA/ FL preference).

Our response: Thanks for these constructive comments. We agree with the opinion that age of seawater will affect DOC/POC quality and microbial community compositions. However, it is not easy to directly connect age of seawater with DOC/POC quality and microbial communities, especially in the case of lacking the measurement and analysis of DOC/POC quality. It is well known that the degree of remineralization and degradation of POC increases as seawater ages. In our study, along vertical depth profiles, the seawater gradually becomes older. Therefore, for POC, older seawater stands for longer sinking distance and higher degradation. To some degree, the

impact of age of seawater to microbial community is similar to that of depth. In our original manuscript, we presented our primary hypothesis to describe this kind of influence from depth (Line 416-424). In response to this comment from the reviewer, we added the following text: "During POC sinking from surface through the water column, and also as seawater ages, the labile organic matter becomes increasingly decomposed, while the more refractory material remains and resists degradation (Simon et al., 2002). In such cases, utilization of the POC in the deep sea by microorganisms depends on the quality and quantity of the remaining POC. Meanwhile, in older seawater, DOC also become more refractory because free-living microorganisms preferentially utilize labile DOC and the remained refarcotory DOC gradually accumulates, which potentially affect microbial community structures."

Figure 9 is not introduced in the results but heavily discussed in the discussion. The results reported for Fig. 9 in the Discussion should be moved to the Results.

Our response: Thanks for this advice and we agree. As stated above, because of the removal of singletons, we adjusted this figure based on the new bacterial OTU table correspondingly (supplementary Table S1). Meanwhile, as suggested by the reviewer, we also moved this figure into the "Results" section as a supplementary material (newly named as Figure S7). Correspondingly, we added one short paragraph to describe this Figure S7 at the end of the subsection of "3.5 Bacterial preference to PA or FL lifestyles" as: "At OTU level, near 1/2 of total OTU numbers (2005 out of 4338 OTUs) are shared by PA and FL fractions (Fig. S7). Phylogenetically, these PA/FL-shared OTUs are mainly fallen into $\alpha$-, $\gamma$-, $\delta$-Proteobacteria, Planctomycetes, Chloroflexi, Bacteroidetes, Marinimicrobia and Acidobacteria. Moreover, taxonomic components of PA/FL-shared OTUs at different levels are primarily similar to those of OTUs retrieved exclusively from PA fractions or FL fractions (Table S1, Fig. S7)."

L602- Bchl a is introduced for the first time with no context on what this is or what it is short for.

[Figure]

Our response: Thanks for pointing out this. We used the full name "bacteriochlorophyll a" to replace the abbreviated "Bchl a".

Technical Corrections:

L37- A high proportion "of" overlap.

Our response: Done.

L140- What is CTAB?

Our response: CTAB is the abbreviation of "hexadecyl trimethyl ammonium bromide". In our revised manuscript, we provided the full name of CTAB like "1% hexadecyl trimethyl ammonium bromide (CTAB)."

L151- "each DNA was" should be each "DNA pellet was"?

Our response: Done.

L259- I am not sure what is meant by incompact.

Our response: I am sorry for this unclear statement. We deleted the word of "incompact".

L388: "were supposed to" is a misleading phase. It sounds like an expectation of a result. Perhaps this would better be "several environmental parameters played a pivotal role...".

Our response: Done.

L403 impaction should be impact.

Our response: Thanks and done.

L412, "It is considered..." I am not sure what the 'subset' is and I think this can be better phrased.

Our response: Yes, we agree. We reworded this sentence as following: "DO is considered as one of the most crucial environmental variables for shaping the compositions of particle-attached bacterial assemblages (Salazar et al., 2016)."

L414-415- should be 'A recent study' (not 'A most recent study').

Our response: Done.

L425 – should be 'unexpected' rather than 'out of our expectation'.

Our response: Agree and done.

L425-426 – should be 'generally exhibits N- or P-limited phytoplankton production'.

Our response: Done.

L436- 'niches' is not the correct word here. Maybe habitats? Locations?

Our response: Yes, agree. We replaced "niches" here with "habitats" as suggested by the reviewer.

L445- The phrase 'significantly divergent' implies statistical significance, but no such test was done to prove that PA and FL communities were significantly different (also in lines 641, 27, and 103). I think just 'divergent' would be acceptable unless a test is incorporated.

Our response: Totally agree. During our revision, we performed the MPPR, ANOSIM and PERMANOVA statistical analyses (Table S3). The results confirm the significant differences with P values <0.05. Therefore, we kept these words.

L463- 'dominantly govern' should just be 'dominate'.

Our response: Done.

L498 – I don't understand the meaning of this phrase: 'nothing is available to elaborate the selection better PA and. . .' I think it needs to be reworded.

Our response: Yes, we agree. We reworded this sentence as like: "However, due to

lack of necessary pure culture or their genome information, it is not yet possible to elaborate their preferences for PA and FL lifestyles."

L580- The phrase 'intelligibly convinced' is unclear. Also the entire sentence L580-583 is a run-on sentence with some unclear phrasing and I'm not sure what the intended meaning is.

Our response: We thank the reviewer for pointing out these problems. We reworded our sentences and corrected the grammar errors. The revised sentences are as below: "Their preference to particle-attached lifestyle in the water column is intelligible. Within normal water column, seawater is usually oxic in spite of low oxygen concentrations. Only on or inside the organic particles where heterotrophic microbes attach and digest organic matter using oxygen as electron acceptor, local anoxic niches are developed with the gradual exhaustion of ambient oxygen, and become suitable for the survival of anaerobic methanogens."

Please also note the supplement to this comment:
https://bg.copernicus.org/preprints/bg-2020-115/bg-2020-115-AC3-supplement.zip

————————————————————

[Figure]

**Fig. 1.** Bacterial and archaeal cell abundances in seawaters at different depths from G3 station and J5 station in the South China Sea, estimated from 16S rRNA gene copy abundances.

[Figure]

Shannon's H

Depth(m)

(a) bacteria

- ▲— G3 PA
- △·· G3 FL
- ■— J5 PA
- □·· J5 FL

Shannon's H

Depth(m)

(b) archaea

**Fig. 2.** Shannon's diversity index calculated for all bacterial and archaeal communities of seawaters collected from G3 station and J5 station in the South China Sea.

[Figure]

**Fig. 3.** Results of PCoA analysis for particle-attached and free-living microbial fractions collected from seawater columns of the South China Sea. (a) PA and FL bacteria; (b) PA and FL archaea.

[Figure]

**Fig. 4.** Results of CCA analysis to correlate several environmental factors including POC, seawater age, salinity and depth to PA and FL microbial communities collected from seawater columns of the SCS.

[Figure]

**Fig. 5.** Taxonomic compositions of particle-attached and free-living bacterial communities of seawaters at different depths along two different water columns in the South China Sea. (a) G3 ; (b) J5.

**Fig. 6.** The relative abundances of families in PA and FL bacterial communities. Dark grey bubbles are for the PA fraction, while light grey bubbles are for the FL fraction.

[Figure]

**Fig. 7.** Taxonomic compositions of particle-attached and free-living archaeal communities of seawaters at different depths along two different water columns in the South China Sea. (a) G3 ; (b) J5.

[Figure]

**Fig. 8.** Odds ratio for each of the families with relatively abundant proportions in each sample.

---

## Author Comment (AC4) · 9 Sep 2020

This study focused on the depth profiles of free-living (FL) and particle-attached (PA) prokaryotes (Bacteria and Archaea) in two sites in the South China Sea (SCS). As of now, there is a few studies to reveal the particulate-attached prokaryotic community structures (especially, about Archaea). 16S rRNA gene deep-sequencing analyses revealed the shift of bacterial and archaeal community structures among different depths. Also, several environmental factors such as depth, seawater age, salinity, POC, DOC, DO, and silicate could be critical for determining the community structures. Phylogenetic analyses revealed that several lineages including alpha-, gammaproteobacteria, Actinobacteria, Bacteroidetes...etc. were overlapped between PA and FL fractions. However, there were differences at family level among them. According to these data, the authors discussed about ecological and biogeochemical roles of FL and PA prokaryotes in the SCS.

Major comments The manuscript is well written. But main limitation is a weak of biogeoscientific discussion of FL and PA (especially PA) fractions (e.g. interaction between chemical composition or degradation of POM, and PA bacteria or archaea) (1). In addition, the critical problem is potential primer biases (especially bacterial primer). The selection of primer set is very important for evaluating the prokaryotic community structure and diversity. Especially, SAR11 clade affiliated with Alphaprotoebacteria seems to be underestimated in this study. This clade is known to be dominant lineage in the oceanic environments, and generally accounted for 15_30% of total prokaryotic cells (Morris et al., 2002 Nature 420: p806-810). Different primer set create different results (e.g. Sanchez et al., 2009, Aquat Microb Ecol, 54: p211-216; Apprill et al., 2015, Aquat Microb Ecol, 75: p129-137) on the community analysis in the ocean (at least Bacterial community analysis). The authors should mention these problems in the discussion section (2). Provide more information on the choice of sites and depths for this work. Moreover, provide more detail profiles of environmental factors collected by a Sea-Bird CTD system (at least seawater temperature, salinity and DO) (3). L152-155: Why did the authors choose these primer set (especially, 27F-533R for Bacteria)? I think the SAR11 clade affiliated with Alphaproteobacteria were underestimated (approximately 15_30% of total 16S rRNA sequences, in general). The selection of primer is one of the most critical factors for evaluating prokaryotic community structures and diversity (4). Provide the data for sequence depth (e.g. rarefaction curves) of 16S rRNA gene used in this study (5).

Response to comment (1): Thanks for this comment. As we responded above (1st reviewer), in-depth discussion on the biogeochemical significance of these finding is

not waraanted due to lack of chemistry data (e.g., composition of POM and DOM). We agree in that our manuscript is mainly focused on the microbiological part and the role of microbes in marine carbon cycle. On the "biogeochemical" part, we focus our discussion on the role of PAM and FLM in oceanic carbon cycling processes, i.e., decomposition of POM, inter-conversion between POM and DOM, and degradation of DOM. To this end, the present study is an extension of our previous work, focusing on both microbiological and biogeochemical aspects of PAM and FLM and their potentials in mediating carbon cycling processes in the ocean. Therefore, revealing the microbial taxa in PA and FL assemblages and profiling variations of their abundance and diversity along the water column provides a foundation for a better understanding of the coupled microbiological and biogeochemical processes in marine carbon cycle. Conceptually we can make some inferences based on the current dataset and findings from previous studies. In the manuscript, we had/added additional discussion on microbial metabolic potential in utilizing certain organic compounds. For examples: "They often maintain . . . . . ., and are capable of degrading high-molecular-weight (HMW) organic compounds. . . . . .." "It is further revealed that PA microbes . . . . . . metabolic and regulatory capabilities of utilizing compositionally varied organic matter, while . . . . . ." "These $\gamma$-proteobacterial members are . . . . . . they are believed to have the abilities to degrade/utilize HMW organic compounds with higher nutrient requirements." "Further phylogenetic assignment revealed . . . . . . belong to the genus Methylobacterium which are strictly aerobic, facultatively methylotrophic bacteria, and can grow on a wide range of carbon compounds." "Genomic information underlines that although these clades have a flexible metabolism utilizing multiple hydrocarbon compounds. . . . . .." "The majority of . . . . . ., and commonly possess the ability to hydrolyze and utilize complex carbon sources. Although their abundance . . . . . . because of their high specificity for organics." "Sva0996 marine group . . . . . . have the ability to assimilate phytoplankton-derived dissolved protein." . . . . . . . . . . .

Response to comment (3): As we responded to the 2nd reviewer, this present work is motivated by our early works (Li et al., 2015) in which some preliminary findings indi-
cated that depth probably exert an impact in structuring microbial assemblages. Therefore, in our present research, we selected two stations in the central basin of the SCS with depths >4,000 m to take the samples and test our hypothesis. One sentence was added in "2.1 Sample collection and environmental parameter measurements" subsection to introduce this background: "Both stations have depth > 4,000 m, providing us the bathyal environments to vertically profile the variation of microbial assemblages with depth."

As for the profiles of environmental factors, we also totally agree. As we described in M&M section, a CTD profiler was used to obtain basic environmental parameters of the water column, including depth, salinity, temperature, and dissolved oxygen (DO) were obtained in situ using a DO sensor integrated in the CTD profiler during the sampling. However, unforturnately, it is a pity that we had not the access to get all the continuous datasets of these fundamental environmental parameters at that time. Therefore, as we presented in our manucript, only those data of our sampling depths were provided.

Response to comment (2) and (4):Thanks for this suggestion and we agree. In our manuscript, we selected the primer sets 27F/533R, targeting the hypervariable V1-V3 regions of 16S rRNA gene which is widely used in bacterial community analysis based on the 454 pyrosequencing (for example, Sun et al., 2014, PLOS one; Fonseca et al., 2019, Front Microbiol). As pointed out by the reviewer and previous studies, it has been demonstrated that the relative abundance of SAR11 clade in seawater could be potentially biased by different primer sets. Therefore, we discussed this kind of possibility of the underestimation of SAR11 clade in our samples as below: "In addition, the percentages of SAR11 clade revealed here seem to be relatively lower compared with those reported in previous studies where the SAR11 clade typically makes up 20 to 40% of the bacterioplankton (Morris et al., 2002; Aprill et al., 2015). It may be related to the sequencing primers used which potentially cause underestimation of SAR11 clade and bias the interpretation of their relative abundances (Aprill et al., 2015)."

Response to comment (5): Done. We provided the rarefaction curves in the supplementary materials named as Figure S1.

Specific comments

L96: Marine Group III (MGII)!Marine Group (MGIII)

Our response: Thanks for pointing out this error. It was a typo and has been corrected.

L150: what amount of template DNA (ng) did author used? And provide the information about DNA concentration (or amount) after DNA extraction.

Our response: Thanks for pointing out these questions. For the PCR amplification, $\sim$10 ng DNA template was used. For DNA extraction, we obtained about 4.48 $\sim$ 29.1 ng/$\mu$l DNA concentration dissolved in $\sim$ 50 ul sterilized deionized water. We provide these information in this section.

L152-155: provide the references of these primers.

Our response: Done. Ohene-Adjei et al., 2007 and Sun et al., 2014 were provided after these primers.

L161: provide the reference or URL for QIIME 1.9.1 software

Our response: Done. Caporaso et al., 2010 was added here.

L177: provide the reference or URL for R packages.

Our response: Done. The URL (https://CRAN.R-project.org/package=vegan) was provided for R packages.

L251 and other lines: If the author described "significantly differences", provide the information about R or Rho values, and P value. Maybe, the ANOSIM or PERMANOVA analyses should be need to clarify statistical differences among communities.

Our response: Thanks for this suggestion and we agree. So, as suggested by the reviewer, during our revision, the statistical analyses including MRPP, ANOSIM and PERMANOVA were performed to clarify the statistical significances. The statistical

results were provided as supplementary materials (see Table S3). All the P values <0.05, indicating statistically significant difference. We reworded this sentence as below: "PCoA analysis revealed that there were significant differences (P values <0.05, Table S3) in bacteria and archaea community structures over the depth profiles and between the FL and PA fractions."

L379 "taxonomically" : add information about the taxonomic levels after this word (e.g. taxonomically (at least family or order?? level)).

Our response: Thanks for this comment. Here we just meant to indicate a potential difference in microbial community compositions. The difference can occur at any level of taxonomy. To avoid the unclear statement, we deleted the word of "taxonomically".

L386 "depth": I think it is better to correct "hydrological condition (e.g. depth)".

Our response: Agree and done.

L413: provide the R or Rho value before P value (R or Rho=????, P>0.05).

Our response: Thanks and done. As shown in Table S3, Mantel test was used to test the statistical significance of environmental factors with microbial compositions. In Table S3, R values and P values were listed. Therefore, here we referred this place to Table S3 as following: "However, POC concentration in the present study is not statistically significantly correlated with either bacterial or archaeal community abundances (P values >0.05) (Table S3)."

L417: I can not understand "utilization of refractory POC by microorganisms depends on the quality of POC". I recognize "refractory" is not usable for microorganisms. "Refractory POC" means "POC in the deep sea"?

Our response: Thanks for pointing out this and the reviewer is right. We now have reworded this "refractory POC" as "POC in the deep sea".

L442-443: -proteobacterial (change italic to regular)

Our response: Done.

L446-449: Again, primer selection is one of the critical factors for evaluating the community composition and diversity. Thus, the authors should add the discussion about primer biases.

Our response: Totally agree. However, I think it would be better if we disscuss this at the end of this paragraph. Because SAR11 clade mainly contributes to the FL bacterial fraction rather than PA fraction. Therefore, at the end of this paragraph, we added several sentences to discuss the potential underestimation about SAR11 clade caused by the primer sets used in our study: "In addition, the percentages of SAR11 clade revealed here seem to be relatively lower compared with those reported in previous studies where the SAR11 clade typically makes up 20 to 40% of the bacterioplankton (Morris et al., 2002; Aprill et al., 2015). It may be related to the sequencing primers used which potentially cause underestimation of SAR11 clade and bias the interpretation of their relative abundances (Aprill et al., 2015)."

L462: What is (li)? Reference?

Our response: Typo and corrected. Here should be a reference, Gong et al., 2012.

L522-523 "statistical analysis": provide the R or Rho, and P values.

Our response:Thanks and done. We added here a referring to Table S3 in which the R values and P values were listed by three different statistical analysis including MPPR, ANOSIM and PERMANOVA.

Figure 3: the authors circled the points (triangles, rhombus+hexagon+star: : :etc.) for representing different clusters in bacterial and archaeal fractions. Are there statistically significant? Provide the results of statistical analyses (and show R or Rho value, and P value).

Our response: Thanks for this advice. Yes, they are statistically significant with P values <0.05 (Table S3). We added one more sentence in the caption of this figure:

"Statistical analyses supported the groups with statistical significances (Table S3)."

Figure 6: x-axis is confused. It is better to delete minor scale marks (e.g. those between 50m and 200m).

Our response: Done! During our this revision, we redrew this figure based on those of dominant families with >3% proportions and adjusted the x-axis and scale marks.

Figure 7, Figure S5: provide the information about failed samples in the legend.

Our response: Thanks and we did this. We added one sentence in the legends of Figure 7 and Figure S5 like: "PA-3000 at G3 station and PA-4000 at J5 station indicate the samples failed in the sequencing of archaeal 16S rRNA gene."

Please also note the supplement to this comment:
https://bg.copernicus.org/preprints/bg-2020-115/bg-2020-115-AC4-supplement.zip

[Figure]

[Figure]

Log(cells/L)

Log(cells/L)

**Depth(m)**

**Depth(m)**

▲ PA-G3
△ FL-G3
■ PA-J5
□ FL-J5

(a) bacteria

(b) archaea

**Fig. 1.** Bacterial and archaeal cell abundances in seawaters at different depths from G3 station and J5 station in the South China Sea, estimated from 16S rRNA gene copy abundances.

[Figure]

Shannon's H

(a) bacteria

(b) archaea

**Fig. 2.** Shannon's diversity index calculated for all bacterial and archaeal communities of seawaters collected from G3 station and J5 station in the South China Sea.

[Figure]

**Fig. 3.** Results of PCoA analysis for particle-attached and free-living microbial fractions collected from seawater columns of the South China Sea. (a) PA and FL bacteria; (b) PA and FL archaea.

[Figure]

**Fig. 4.** Results of CCA analysis to correlate several environmental factors including POC, seawater age, salinity and depth to PA and FL microbial communities collected from seawater columns of the SCS.

[Figure]

**Fig. 5.** Taxonomic compositions of particle-attached and free-living bacterial communities of seawaters at different depths along two different water columns in the South China Sea. (a) G3 ; (b) J5.

**Fig. 6.** The relative abundances of families in PA and FL bacterial communities. Dark grey bubbles are for the PA fraction, while light grey bubbles are for the FL fraction.

[Figure]

**Fig. 7.** Taxonomic compositions of particle-attached and free-living archaeal communities of seawaters at different depths along two different water columns in the South China Sea. (a) G3 ; (b) J5.

Fig. 8. Odds ratio for each of the families with relatively abundant proportions in each sample.

---

## Author Response (AR2)

Dear Dr. Robinson,

We are pleased to know that our manuscript will be accepted after a technical correction. We agree that, as suggested by one of the reviewers, salinity is a dimensionless entity. Therefore, "PSU" after the salinity values between Line 234 and Line 235 should be deleted. Thank you again for handling our manuscript.

Best regards,

Jiasong Fang